# Selective Generation for Controllable Language Models

**Minjae Lee**\*
GSAI
POSTECH
minjae.lee@postech.ac.kr

**Kyungmin Kim**\*
GSAI
POSTECH
kkm959595@postech.ac.kr

**Taesoo Kim**
SCS & SCP
GaTech
taesoo@gatech.edu

**Sangdon Park**
GSAI & CSE
POSTECH
sangdon@postech.ac.kr

## Abstract

Trustworthiness of generative language models (GLMs) is crucial in their deployment to critical decision making systems. Hence, certified risk control methods such as selective prediction and conformal prediction have been applied to mitigating the hallucination problem in various supervised downstream tasks. However, the lack of appropriate correctness metric hinders applying such principled methods to language generation tasks. In this paper, we circumvent this problem by leveraging the concept of *textual entailment* to evaluate the correctness of the generated sequence, and propose two selective generation algorithms which control the false discovery rate with respect to the textual entailment relation (FDR-E) with a theoretical guarantee: $\texttt{SGen}^{\texttt{Sup}}$ and $\texttt{SGen}^{\texttt{Semi}}$. $\texttt{SGen}^{\texttt{Sup}}$, a direct modification of the selective prediction, is a supervised learning algorithm which exploits entailment-labeled data, annotated by humans. Since human annotation is costly, we further propose a semi-supervised version, $\texttt{SGen}^{\texttt{Semi}}$, which fully utilizes the unlabeled data by pseudo-labeling, leveraging an *entailment set function* learned via conformal prediction. Furthermore, $\texttt{SGen}^{\texttt{Semi}}$ enables to use more general class of selection functions, *neuro-selection functions*, and provides users with an optimal selection function class given multiple candidates. Finally, we demonstrate the efficacy of the $\texttt{SGen}$ family in achieving a desired FDR-E level with comparable selection efficiency to those from baselines on both open and closed source GLMs. Code and datasets are provided at `https://github.com/ml-postech/selective-generation`.

## 1 Introduction

Generative language models (GLMs) [1, 2, 3, 4] have garnered significant attention for their ability to generate human-level language [5] primarily due to underlying transformer architectures [6]. However, GLMs raise concerns about generating hallucinated facts [7], which is an undesirable property when they are used as knowledge retrieval sources. This issue can be mitigated by fine-tuning with human feedback [7, 8], but it remains expensive in terms of training and labeling costs. Certified risk control methods such as selective prediction [9] and conformal prediction [10] are promising cost-efficient alternatives, which have been applied to the hallucination mitigation in various supervised downstream tasks [9, 10, 11, 12, 13, 14].

---

\*Equal contribution

38th Conference on Neural Information Processing Systems (NeurIPS 2024).

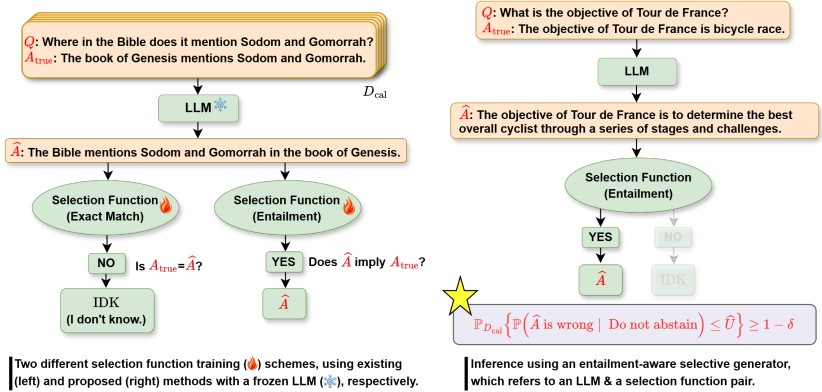

Figure 1: An overview and qualitative results of our method with GPT-3.5-Turbo. The crux is to learn an entailment-aware selective generator with an abstaining option that controls the rate of hallucination (in a false discovery rate) over generated sequences with a probabilistic guarantee.

The main bottleneck in applying such certified methods to language generation tasks is that provided risk control guarantees require correctness labels during the learning process. Specifically, in classification, high-quality correctness labels can be directly acquired by comparing true and predicted labels using exact match (EM). However, this is not the case for language generation tasks, since multiple valid answers can exist for the same question. As correctness metrics such as EM and F1-score do not account for the multiple valid answers, directly applying them to language generation tasks results in a significant gap between the true and measured correctness, which we call the *metric misalignment*. Thus, a correctness evaluation metric that accounts for multiple answers is required.

In this paper, we resolve the metric misalignment problem by leveraging *textual entailment* to evaluate the correctness of generated answers and define the false discovery rate with respect to the textual entailment relation (FDR-E). Given two ordered sequences, a premise and a hypothesis, we say that the premise entails the hypothesis if the hypothesis is true given the premise. Based on this notion of entailment, we propose two selective generation algorithms, $\mathtt{SGen}^{\mathtt{Sup}}$ and $\mathtt{SGen}^{\mathtt{Semi}}$, which are generalized versions of selective classification [9] to control the FDR-E by abstaining from returning an answer when a GLM is uncertain of its answer.

In particular, $\mathtt{SGen}^{\mathtt{Sup}}$, a direct modification of [9], is a supervised selective generator learning algorithm which requires entailment labels. This necessitates human annotations on textual entailment, where a generated answer is the premise and a true answer is the hypothesis. As labeling is expensive and $\mathtt{SGen}^{\mathtt{Sup}}$ solely relies on entailment-labeled data, we propose a semi-supervised method, $\mathtt{SGen}^{\mathtt{Semi}}$, which enables the exploitation of entailment-unlabeled data in learning a selective generator by pseudo-labeling textual entailment using an *entailment set function* learned via conformal prediction [10]. Based on an entailment classifier originally developed for the natural language inference problem [15, 16], the estimated entailment set function approximates a true entailment set function, which returns all entailed answers if a true answer is given as a hypothesis.

Additionally, $\mathtt{SGen}^{\mathtt{Semi}}$ introduces the general class of selection functions for selective generation, called *neuro-selection functions*. In selective prediction, learning a selective predictor is equivalent to learning a selection function, which is an indicator function to decide whether to abstain from returning a prediction. The standard selective prediction algorithm [9] considers the class of single-threshold indicator functions using a pre-specified confidence-rate function. For the same risk level, the better the confidence-rate function quantifies the model's uncertainty, the less likely the selective predictor is to abstain from making a prediction. We refer to this as *selection efficiency* henceforth. As appropriate confidence calibration for language generation remains challenging, optimizing a single-threshold indicator function with a poorly calibrated confidence-rate function leads to low selection efficiency. Instead, we generalize the selection function by using a multiple-threshold indicator function with trainable features. Furthermore, $\mathtt{SGen}^{\mathtt{Semi}}$ provides a user with an optimal class of selection functions among possible candidates in terms of the FDR-E.

Finally, we empirically demonstrate the efficacy of $\mathtt{SGen}^{\mathtt{Semi}}$ over open and closed source GLMs, where we consider $\mathtt{SGen}^{\mathtt{Sup}}$ as one of our baselines as it is a direct modification of [9]. To validate

our method and its theoretical guarantee, we create a new dataset on textual entailment using the Natural Questions (NQ) dataset [17] for each GLM. Given a question and answer pair, the textual entailment is labeled by letting a generated answer as a premise and the true answer in declarative form as a hypothesis. As communities lack human-annotated entailment-labeled data for language generation, we believe that our dataset contributes to the hallucination evaluation of GLMs. For both open and closed source GLMs, `SGen`[Semi] is effective in achieving a desired FDR-E level with better selection efficiency compared to baselines.

## 1.1 Related Work

We introduce two main research directions to mitigate hallucination in GLMs.

**Heuristics for hallucination mitigation.** The hallucination in language generation usually refers to the situation where a GLM generates wrong answers with high confidence, which hinders the reliable deployment of GLMs. As fine-tuning methods are expensive, heuristics for hallucination mitigation without tuning have been proposed [18, 19]. Notably, [19] proposes a performant hallucination detection method, which quantifies the self-consistency among multiple generated answers for the same question using textual entailment models to detect the hallucination. However, these methods do not provide certified control over the occurrence of hallucinated contents.

**Certified methods for hallucination mitigation.** Conformal prediction outputs a prediction set that is guaranteed to contain a true label with high probability, where a provided coverage guarantee is model-agnostic under a mild assumption on a data [10]. Although this property enables the safe deployment of complex models and has made conformal prediction popular [10, 12, 13, 20, 21, 22], the constructed prediction sets in language generation are often less-informative due to an unbounded label space, which frequently renders the coverage guarantee ineffective [23, 24]. To restrict the prediction set to a moderate size, [23] constructs the prediction set over answers by sampling them sequentially, while still satisfying the coverage guarantee. Still, post-selection of answers from the prediction set is necessary for final decision making, which may result in the selection bias [25, 26]. [27, 28] decompose generated answers into alignment-labeled sub-claims and return a set of sub-claims that contains no contradiction with high probability via conformal prediction. Even though the post-selection is unnecessary, it requires expensive alignment labels for every sub-claim.

Unlike conformal prediction, selective prediction directly manages target risk at a desired level by introducing an abstaining option on unsure predictions. [9] proposes a selective prediction method mainly for classification, which learns a threshold-based selection function that controls the false discovery rate (FDR) to a desired level. [24] generalizes the selective prediction to language generation. However, their theoretical guarantee is not focused on the target risk to control, but on a consistency property of a surrogate loss function with respect to a true loss function in optimization process. [29], concurrently published with our paper, proposes a certified selective generation method for context-given language generation which controls the FDR. Unlike [9] which takes the number of selected samples as constraint in learning the selection function, [29] set the power as constraint. However, as [24] does, they require an additional calibration set for training an entailment scoring function. Importantly, while existing selective generation methods are supervised learning methods, we propose a semi-supervised learning algorithm that can fully leverage entailment-unlabeled data.

## 2 Background

While we consider general language generation tasks, we confine our scope to the open-ended question-answering task and define the notation accordingly for the sake of clarity and for maintaining consistency in descriptions on the experiment. Specifically, let $\mathcal{W}$ denote a token space constructed using a tokenizer, such as Byte Pair Encoding [30], and let $\mathcal{W}^*$ denote a token sequence space, defined as $\mathcal{W}^* := \cup_{i=0}^{\infty} \mathcal{W}^i$. Let $(\mathbf{x}, \mathbf{y}) \in \mathcal{X} \times \mathcal{Y}$ be a question and answer sequence pair, where $\mathcal{X} := \mathcal{W}^*$ and $\mathcal{Y} := \mathcal{W}^*$ refer to the token sequence spaces of questions and answers, respectively. We assume the answer sequence is in a declarative form. Finally, $\mathbf{x}_{i:j}$ refers to the sub-sequence of $\mathbf{x}$ from the $i$-th to the $j$-th token.

### 2.1 Language Generation

Given a question as input, a GLM generates an answer through the sequential process called decoding, which we call language generation. Here, we consider the greedy decoding, a deterministic generation process described as follows. Let $p_M : \mathcal{X} \times \mathcal{W} \to \mathbb{R}_{\geq 0}$ denote a GLM which returns a next-token

distribution given the input sequence $\mathbf{x}$, where $\sum_{w \in \mathcal{W}} p_M(w \mid \mathbf{x}) = 1$ for all $\mathbf{x} \in \mathcal{X}$. A language generator $G : \mathcal{X} \to \mathcal{Y}$ using greedy decoding sequentially generates tokens from the GLM as follows: $\hat{\mathbf{y}}_i := \arg\max_{w \in \mathcal{W}} p_M(w \mid (\mathbf{x}, \hat{\mathbf{y}}_{1:i-1}))$ for $i \geq 2$ and $\hat{\mathbf{y}}_1 := \arg\max_{w \in \mathcal{W}} p_M(w \mid \mathbf{x})$. The generator $G$ returns a generated answer $\hat{\mathbf{y}} := G(\mathbf{x})$ and terminates the decoding process when the end-of-sequence (EOS) token is returned. Here, the conditional probability of the answer $\hat{\mathbf{y}}$ is defined as $f_M(\mathbf{x}, \hat{\mathbf{y}}) := p_M(\hat{\mathbf{y}}_1 \mid \mathbf{x}) \prod_{i=2}^{|\hat{\mathbf{y}}|} p_M(\hat{\mathbf{y}}_i \mid (\mathbf{x}, \hat{\mathbf{y}}_{1:i-1}))$, commonly used as its uncertainty measure.

## 2.2 Selective Prediction

Selective prediction refuses to make a prediction by returning "I don't know" (IDK) if the prediction is uncertain. In classification, the selective classifier $\hat{S}$ consists of a pair of a classifier $\hat{y}$ and a selection function $\hat{s}$, and is defined as follows: $\hat{S}(\mathbf{x}) := \begin{cases} G(\mathbf{x}) & \text{if } \hat{s}(\mathbf{x}) = 1 \\ \text{IDK} & \text{otherwise} \end{cases}$, where $\hat{y}(\mathbf{x}) := \arg\max_{y \in \mathcal{Y}} f(\mathbf{x}, y)$. Here, $f(\mathbf{x}, y)$ refers to an estimated likelihood of the given input $\mathbf{x}$ for being a class $y$, determined by an underlying classification model $f$. Although the selection function can be of arbitrary form, the common choice is a single threshold indicator function using the maximum likelihood as the confidence-rate function, *i.e.,* $\hat{s}(\mathbf{x}) := \mathbb{1}(f(\mathbf{x}, \hat{y}) \geq \tau)$. Here, the confidence-rate function is defined to quantify the uncertainty of the model's prediction. Under the independent and identically distributed (i.i.d.) assumption, [9] proposed the certified threshold learning algorithm which controls the false discovery rate (FDR) with respect to the EM metric with the PAC guarantee, where the FDR is defined as $\mathcal{R}_{\text{EM}}(\hat{S}) := \mathbb{P}\{\hat{y}(\mathbf{x}) \neq y \mid \hat{S}(\mathbf{x}) \neq \text{IDK}\}$. Since EM considers the answer $\hat{y}(\mathbf{x})$ to be correct when it is exactly the same as the reference answer $y$, it is an inappropriate correctness metric for language generation problems that can have multiple valid sequences for the same input. This results in learning a too conservative and vacuous selection function for language generation, which is empirically verified by our experiments. Thus, we leverage the textual entailment to evaluate the correctness of the generated sequence to alleviate the metric misalignment problem.

## 2.3 Textual Entailment

Natural language inference (NLI), also denoted as recognizing textual entailment, predicts whether one sequence implies another. The former refers to a premise ($\mathbf{p}$), and the latter refers to a hypothesis ($\mathbf{h}$). Since the release of two large-scale benchmarks of ordered sequence pairs labeled with textual entailment [15, 16], a number of transformer-based entailment classifiers have been proposed and shown impressive results. Each pair is classified into one of three categories: *entailment* if $\mathbf{h}$ is true given $\mathbf{p}$; *contradiction* if $\mathbf{h}$ is false given $\mathbf{p}$; and *neutral* otherwise. In this paper, we define the entailment scoring function as $f_E(G(\mathbf{x}), \mathbf{y}) := 1 - p_E(contradict \mid \mathbf{p} = G(\mathbf{x}), \mathbf{h} = \mathbf{y})$ to estimate and pseudo-label the correctness of $G(\mathbf{x})$, where $p_E(contradict \mid \mathbf{p} = G(\mathbf{x}), \mathbf{h} = \mathbf{y})$ is the likelihood that $G(\mathbf{x})$ contradicts $\mathbf{y}$. While pseudo-labeling enables the full exploitation of unlabeled data to learn a selection function, controlling the mislabeling error remains as a challenge.

## 2.4 Conformal Prediction

Conformal prediction [10] outputs a prediction set to quantify the uncertainty of a given model with a model-agnostic correctness guarantee under minimal assumptions on data generating process. Specifically, under the i.i.d. assumption, PAC conformal prediction [11] incorporates the interpretation of tolerance regions [31] and training-conditional inductive conformal prediction [20] through the lens of PAC learning theory [32]. In this paper, we adopt the PAC prediction set learning algorithm to control the rate of mislabeling error in pseudo-labeled samples used to learn a selection function for selective generation. See Section A.1 for detailed discussion on conformal prediction.

**Scalar-parameterized Conformal Set.** In this paper, we consider a conformal set $C : \mathcal{X} \to 2^{\mathcal{Y}}$ parameterized by a scalar [11, 33] as $C(\mathbf{x}) := \{y \in \mathcal{Y} \mid f(\mathbf{x}, y) \geq \tau\}$, where $\tau \in \mathcal{H}$ is a scalar parameter to learn, $\mathcal{H}$ is a hypothesis space (*e.g.,* $\mathcal{H}$ a finely discretized non-negative real numbers), and $f : \mathcal{X} \times \mathcal{Y} \to \mathbb{R}_{\geq 0}$ is called a *scoring function*. The scoring function corresponds to a target model whose uncertainty is to be quantified, where the softmax output is a common choice in classification. Specifically, $f(\mathbf{x}, y)$ measures the likelihood of $y$ as a response given $\mathbf{x}$ as input.

**PAC Guarantee.** The PAC prediction set learning algorithm outputs a conformal set $\hat{C}$ which upper bounds a miscoverage rate $\mathcal{R}_{\text{MC}}(\hat{C}) := \mathbb{P}\{y \notin \hat{C}(\mathbf{x})\}$ to a desired level $\varepsilon \in (0, 1)$, where the miscoverage rate can be generalized to risk $\mathcal{R}_{01}(\hat{C}) := \mathbb{E}\{\ell_{01}(\hat{C}, \mathbf{x}, y)\}$, on any indicator losses that are monotonic with respect to $\tau$. The algorithm is *probably approximately correct* (PAC) in

the sense that it provides a calibration data-conditional guarantee at every risk and confidence level. Specifically, it controls the risk to a desired level irrespective of which calibration data is used to learn $\hat{C}$ with a desired confidence $\delta \in (0,1)$ as follows: $\mathbb{P}\{\mathcal{R}_{01}(\hat{C}) \leq \varepsilon\} \geq 1 - \delta$, where the probability is taken over the calibration set $\mathbf{Z} \sim \mathcal{D}^n$ to learn the conformal set. In this paper, we leverage the PAC conformal set for a pseudo-labeling function such that the guarantee on the labeling quality provides the overall PAC guarantee in semi-supervised selective generator learning algorithm.

**Algorithm.** The PAC conformal set learning algorithm $\mathcal{A}_{\text{Binom}} : (\mathcal{X} \times \mathcal{Y})^* \to \mathcal{H}$ [11, 20, 34] returns the conformal set parameter $\hat{\tau}$, where $\mathcal{H}$ is a finely-discretized $\mathbb{R}_{\geq 0}$. Specifically, the algorithm returns $\hat{\tau} = \max_{\tau \in \mathcal{H}} \tau$ subject to $U_{\text{Binom}}(k_\tau; n, \delta) \leq \varepsilon$, where $k_\tau := \sum_{i=1}^{n} \ell_{01}(\hat{C}, \mathbf{x}_i, y_i)$. Letting $F(k; n, \theta)$ be a cumulative distribution function of a binomial distribution with $n$ trials and success probability $\theta$, $U_{\text{Binom}}(k; n, \delta) := \inf \{\theta \in [0, 1] \mid F(k; n, \theta) \leq \delta\} \cup \{1\}$ is an upper binomial tail bound that satisfies $\mathbb{P}\{\mathcal{R}_{01}(\hat{C}) \leq U_{\text{Binom}}(k_\tau; n, \delta)\} \geq 1 - \delta$, where $\delta$ is the desired confidence. Note that we similarly denote a lower binomial tail bound by $L_{\text{Binom}}$. If optimization in the algorithm $\mathcal{A}_{\text{Binom}}$ is infeasible, the algorithm returns $\hat{\tau} = 0$, a vacuous conformal set. Thus, the algorithm is PAC, and see Section A.1 for proof.

## 2.5 Calibration

In classification, calibration aims to adjust the classifier's maximum likelihood response, or confidence, to be correct. We say the classifier response $f : \mathcal{X} \times \mathcal{Y} \to \mathbb{R}_{>0}$ is *perfectly calibrated* with respect to a distribution $\mathcal{D}$ over $\mathcal{X} \times \mathcal{Y}$ and a classifier $\hat{y}$ if $\mathbb{P}\{\mathbf{y} = \hat{y}(\mathbf{x}) \mid f(\mathbf{x}, \hat{y}(\mathbf{x})) = t\} = t$ for all $t \in [0, 1]$ [35, 36]. Calibration aims to find the classifier response such that it is perfectly calibrated asymptotically. In this paper, we make an interesting connection between calibration and selective generation. In particular, given the definition of the perfect calibration for a language scoring function $f_M$, we formally provide a sufficient condition for a selective generator to control the FDR with respect to the textual entailment relation at *any* desired risk level.

# 3 Problem: Selective Generation

Let $\mathbf{x} \in \mathcal{X}$ be a question and $\mathbf{y} \in \mathcal{Y}$ be an answer, assuming that each question has a desired answer. Here, we assume $(\mathbf{x}, \mathbf{y}) \overset{\text{i.i.d.}}{\sim} \mathcal{D}'$, where $\mathcal{D}'$ is a data generating process of question-answering pairs. Then, given a generator $G : \mathcal{X} \to \mathcal{Y}$, we consider a *selective generator* $\hat{S} : \mathcal{X} \to \mathcal{Y} \cup \{\text{IDK}\}$ which refuses to return $G(\mathbf{x})$ if a selection function $\hat{s}(\mathbf{x}, G(\mathbf{x})) \in \{0, 1\}$ deems uncertain as follows:

$$\hat{S}(\mathbf{x}) := \begin{cases} G(\mathbf{x}) & \text{if } \hat{s}(\mathbf{x}, G(\mathbf{x})) = 1 \\ \text{IDK} & \text{otherwise.} \end{cases}$$

Our main goal is to learn a selective generator $\hat{S}$ to control a generalized false discovery rate (FDR) with respect to a relation $R$ as

$$\mathcal{R}_R(\hat{S}) := \mathbb{P}\left\{(G(\mathbf{x}), \mathbf{y}) \notin R \mid \hat{S}(\mathbf{x}) \neq \text{IDK}\right\}. \tag{1}$$

Here, the probability is taken over examples $(\mathbf{x}, \mathbf{y}, e, v)$, where $e := \mathbb{1}((G(\mathbf{x}), \mathbf{y}) \in R)$ is an additional label to be annotated due to unknown $R$ and $v \in \{0, 1\}$ is a visibility flag of $e$ for semi-supervised learning. For the data generation of $(\mathbf{x}, \mathbf{y}, e, v)$, we assume that a label $e$ is observed with an unknown success probability of $p_v$, independent of the generative process of $(\mathbf{x}, \mathbf{y}, e)$, *i.e.*, $(\mathbf{x}, \mathbf{y}, e, v) \sim \mathcal{D} := \mathcal{D}' \cdot \mathcal{V}$, where $\mathcal{D}'$ is a distribution over $\mathcal{X} \times \mathcal{Y} \times \{0, 1\}$ and $\mathcal{V} := \text{Bernoulli}(p_v)$. Note that the definition of $e$, $\mathcal{D}'$ varies by generator $G$ even with the same data generating distribution of $(\mathbf{x}, \mathbf{y})$. In this paper, we design a learning algorithm $\mathcal{A}$ that returns a selective generator $\hat{S}$ to control the generalized FDR with respect to $R$ within a desired level $\varepsilon \in (0, 1)$ with probability at least $1 - \delta \in (0, 1)$, *i.e.*, $\mathbb{P}\{\mathcal{R}_R(\mathcal{A}(\mathbf{Z})) \leq \varepsilon\} \geq 1 - \delta$. Here, the probability is taken over a calibration set $\mathbf{Z} \sim \mathcal{D}^n$. This guarantee is called a probably approximately correct (PAC) guarantee [32]. Among selective generators that satisfies the PAC guarantee, we choose one that minimizes the ratio of IDK-answers with the highest *selection efficiency*. The main challenge is to find a sample and selection efficient PAC algorithm for any $\varepsilon$ and $\delta$ along with designing a relation $R$ for structured labels, as in question-answering. Frequently, we may not obtain a PAC algorithm for any $\varepsilon$, so in this paper, we use a relaxed notion of *controllable* instead of *correct* if the algorithm provides minimum achievable risk beoyond a given $\varepsilon$.

# 4 Semi-Supervised Learning for Controllable Selective-Generation

In this paper, we leverage the textual entailment as the evaluation metric in language generation to consider multiple valid answers in a principled way, and propose two selective generator learning algorithms which control FDR with respect to the textual entailment: $\mathsf{SGen}^{\mathsf{Sup}}$ and $\mathsf{SGen}^{\mathsf{Semi}}$.

## 4.1 False Discovery Rate via Textual Entailment (FDR-E)

A textual entailment relation $R_E$ is an ordered subset of $\mathcal{Y} \times \mathcal{Y}$ where $(\mathbf{y}', \mathbf{y}) \in R_E$ if $\mathbf{y}'$ entails $\mathbf{y}$. In question-answering as an example, the generated answer $G(\mathbf{x})$ is correct if the reference answer $\mathbf{y}$ is a logical consequence of $G(\mathbf{x})$. In other words, $G(\mathbf{x})$ is valid if $G(\mathbf{x}) \in E_{\text{true}}(\mathbf{y})$, where the true entailment set function $E_{\text{true}} : \mathcal{Y} \to 2^{\mathcal{Y}}$ is defined as follows: $E_{\text{true}}(\mathbf{y}) \coloneqq \{\mathbf{y}' \in \mathcal{Y} \mid (\mathbf{y}', \mathbf{y}) \in R_E\}$. Then, an FDR with respect to the entailment relation $R_E$ (FDR-E) that we aim to control is as follows:

$$\mathcal{R}_{R_E}(\hat{S}) \coloneqq \mathbb{P}\{G(\mathbf{x}) \notin E_{\text{true}}(\mathbf{y}) \mid \hat{S}(\mathbf{x}) \neq \mathtt{IDK}\},$$

where the probability is taken over labeled examples, *i.e.,* $(\mathbf{x}, \mathbf{y}, e) \sim \mathcal{D}$. Here, the label $e$ is specifically called an entailment label, *i.e.,* $e \coloneqq G(\mathbf{x}) \in E_{\text{true}}(\mathbf{y})$. Then, for any $G, \mathcal{D}, \mathcal{V},$ and $\hat{S}$, the FDR-E can be decomposed as follows:

$$\underbrace{\mathbb{P}_{\mathcal{D}_{\hat{S}}}\{G(\mathbf{x}) \notin E_{\text{true}}(\mathbf{y})\}}_{(A)} = \underbrace{\mathbb{P}_{\mathcal{D}_{\hat{S}}}\{v=1\}}_{(B)} \underbrace{\mathbb{P}_{\mathcal{D}_{\hat{S}}}\{e=0\}}_{(C)} + \underbrace{\mathbb{P}_{\mathcal{D}_{\hat{S}}}\{v=0\}}_{(D)} \underbrace{\mathbb{P}_{\mathcal{D}_{\hat{S}}}\{e=0\}}_{(E)}, \qquad (2)$$

where $\mathbb{P}_{\mathcal{D}_{\hat{S}}}\{\cdot\} \coloneqq \mathbb{P}\{\cdot \mid \hat{S}(\mathbf{x}) \neq \mathtt{IDK}\}$. Note that as $(\mathbf{x}, \mathbf{y}, e)$ and $v$ are independent, (A), (C), and (E) in (2) are of the same quantity, which is the target risk that we aim to find an upper bound.

## 4.2 FDR-E Bound for Supervised Learning

We first propose the supervised learning algorithm $\mathsf{SGen}^{\mathsf{Sup}}$ (Algorithm 8), a direct modification of [9] to language generation tasks. In particular, $\mathsf{SGen}^{\mathsf{Sup}}$ is a supervised method in the sense that it solely exploits labeled examples $\mathbf{Z}_E \coloneqq \{(\mathbf{x}, \mathbf{y}, e) \mid (\mathbf{x}, \mathbf{y}, e, v) \in \mathbf{Z} \wedge v = 1\}$ to learn a selective generator that controls the upper bound (C) in (2). Note that for supervised learning, we assume that (B) in (2) is always 1, so we only consider the the upper bound (C) via the binomial tail bound as [9].

## 4.3 FDR-E Bound for Semi-Supervised Learning

As $\mathsf{SGen}^{\mathsf{Sup}}$ requires human annotations for entailment labels and makes no use of abundant unlabeled examples $\mathbf{Z}_U \coloneqq \{(\mathbf{x}, \mathbf{y}) \mid (\mathbf{x}, \mathbf{y}, e, v) \in \mathbf{Z} \wedge v = 0\}$, we further propose a novel semi-supervised learning algorithm $\mathsf{SGen}^{\mathsf{Semi}}$ (Algorithm 5), which fully exploits both $\mathbf{Z}_E$ and $\mathbf{Z}_U$ while controlling the FDR-E in (2). In particular, we (1) estimate a true entailment set $E_{\text{true}}$ via conformal prediction with labeled examples $\mathbf{Z}_E$ and then (2) use the estimated entailment set $\hat{E}$ to annotate pseudo-labels on $\mathbf{Z}_U$. Finally, we (3) use both labeled and pseudo-labeled examples to learn a selective generator. Interestingly, this heuristic-looking algorithm could be a rigorous algorithm that controls the FDR-E of a selective generator, which will be described in the following sections.

### 4.3.1 FDR-E Decomposition

$\mathsf{SGen}^{\mathsf{Semi}}$ leverages unlabeled examples by estimating an entailment set as a pseudo-labeling function. However, the estimation error introduces wrong pseudo-labels. Here, we consider a rigorous way to derive the FDR-E upper bound by controlling the estimation error of the pseudo-labeling function. In particular, two different types of estimation errors of an estimated entailment set $\hat{E}$ are illustrated in Figure 2, *i.e.,* a false negative entailment rate (FNER) and a false entailment rate (FER). This results in the following decomposition.

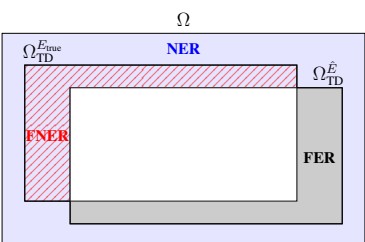

Figure 2: Decomposition of a false discovery rate with respect to an entailment set $E_{\text{true}}$ (FDR-E). Here, $\Omega_{\text{TD}}^E \coloneqq \{(\mathbf{x}, \mathbf{y}, e, v) \mid G(\mathbf{x}) \in E(\mathbf{y})\}$.

**Lemma 1.** *(E) in (2) is decomposed as follows:*

$$\underbrace{\mathbb{P}_{\mathcal{D}_{\hat{S}}}\{e=0\}}_{(E)} = \underbrace{\mathbb{P}_{\mathcal{D}_{\hat{S}}}\{e=0, \hat{e}=1\}}_{FER} - \underbrace{\mathbb{P}_{\mathcal{D}_{\hat{S}}}\{e=1, \hat{e}=0\}}_{FNER} + \underbrace{\mathbb{P}_{\mathcal{D}_{\hat{S}}}\{\hat{e}=0\}}_{NER}. \qquad (3)$$

Here, the first two terms are related to the entailment label estimation error and the last term is the approximate FDR-E using pseudo-labels. As three terms are inter-related, we choose to control the FER term to control (E) in (2) via conformal prediction in the following section.

### 4.3.2 Pseudo-labeling via Conformalized Entailment Set Learning

$\texttt{SGen}^{\texttt{Semi}}$ leverages the PAC conformal prediction for the entailment label estimation to control the mislabeling error. Specifically, we estimate the true entailment set function $E_{\text{true}}$ via an estimated entailment set $\hat{E}$ using $\mathbf{Z}_E$, where we use the entailment scoring function $f_E$ as a scoring function, *i.e.*, $\hat{E}(\mathbf{y}) := \{\mathbf{y}' \in \mathcal{Y} \mid f_E(\mathbf{y}', \mathbf{y}) \geq \tau_E\}$. Here, the corresponding loss $\ell(\hat{E}, \mathbf{x}, \mathbf{y}, e) := \mathbb{1}(e = 0 \land G(\mathbf{x}) \in \hat{E}(\mathbf{y}))$ is a monotonically non-increasing function with respect to $\tau_E$, so we can use the PAC conformal set learning algorithm. Given a desired risk $\varepsilon_E$ and confidence $\delta_E$ level, the corresponding algorithm $\mathcal{A}_{\text{FER}}$ (*i.e.*, Algorithm 1) returns the estimated entailment set function $\hat{E}$ which controls the *false entailment rate* (FER) of pseudo-labeled examples $\mathcal{R}_{\text{FER}}(\hat{E}) := \mathbb{P}_{\mathcal{D}_{\hat{S}}} \{e = 0 \land G(\mathbf{x}) \in \hat{E}(\mathbf{y})\}$ with the following PAC guarantee, where the probability is taken over training examples from $\mathcal{D}_{\hat{S}}$.

$$\mathbb{P}\{\mathcal{R}_{\text{FER}}(\hat{E}) \leq \varepsilon_E\} \geq 1 - \delta_E. \tag{4}$$

### 4.3.3 FDR-E Bound

We then bound the FDR-E for semi-supervised learning, *i.e.*, (E) in (2), via the PAC guarantee by the conformal set learning on $\mathbf{Z}_E$ and the binomial tail bound on $\mathbf{Z}_E$ and $\mathbf{Z}_U$. In particular, the FER is upper-bounded by $\varepsilon_E$, the FNER is lower-bounded by the binomial tail bound using $\mathbf{Z}_E$, and NER is upper-bounded by the binomial tail bound using $\mathbf{Z}_U$. These bounds hold with high probability, and are therefore combined via a union bound, as in the following lemma. See Appendix G for a proof.

**Lemma 2.** *Let* $\hat{\mathbf{Z}}_E := \{(\mathbf{x}, \mathbf{y}, e) \in \mathbf{Z}_E \mid \hat{S}(\mathbf{x}) \neq \texttt{IDK}\}$ *and* $\hat{\mathbf{Z}}_U := \{(\mathbf{x}, \mathbf{y}) \in \mathbf{Z}_U \mid \hat{S}(\mathbf{x}) \neq \texttt{IDK}\}$. *For any* $G$, $\mathcal{D}$, $\mathcal{V}$, *and* $\hat{S}$, *if* $\hat{E} := \mathcal{A}_{\text{FER}}(\hat{\mathbf{Z}}_E)$ *satisfies* $\mathbb{P}_{\hat{\mathbf{Z}}_E}\{\mathcal{R}_{\text{FER}}(\hat{E}) \leq \varepsilon_E\} \geq 1 - \delta'_E/2$, *we have*

$$\mathbb{P}_{\mathcal{D}}\{e = 0\} \leq \varepsilon_E - L_{\text{Binom}}(\hat{k}; |\hat{\mathbf{Z}}_E|, \delta'_E/2) + U_{\text{Binom}}(\hat{l}; |\hat{\mathbf{Z}}_U|, \delta'_S) =: U_{SSL} \tag{5}$$

*with probability at least* $1 - \delta'_E - \delta'_S$, *where the probability is taken over* $\mathbf{Z}$. *Here,* $\hat{k} := \sum_{(\mathbf{x}, \mathbf{y}, e) \in \hat{\mathbf{Z}}_E} \mathbb{1}(e = 1 \land G(\mathbf{x}) \notin \hat{E}(\mathbf{y}))$ *and* $\hat{l} := \sum_{(\mathbf{x}, \mathbf{y}) \in \hat{\mathbf{Z}}_U} \mathbb{1}(G(\mathbf{x}) \notin \hat{E}(\mathbf{y}))$.

Notably, each of three bounds holds over a conditional distribution $\mathcal{D}_{\hat{S}}$, but Lemma 2 relaxes this to an unconditional distribution $\mathcal{D}$ for our final FDR-E guarantee.

**Optimizing the FDR-E Bound (5).** Lemma 2 introduces a hyper-parameter $\varepsilon_E$, which controls a trade-off between the FER and other terms. To find a best trade-off, we optimize $\varepsilon_E$ to minimize the upper bound (5) among $Q$ candidates of $\varepsilon_E$ via $\mathcal{A}_{U_{SSL}\text{-Opt}}$, described in Algorithm 3. This optimization algorithm can find a tighter FDR-E bound, as in the following lemma. See Appendix H for a proof.

**Lemma 3.** *Let* $U_{SSL}$ *be as in (5) and* $\mathcal{Q}$ *be the* $Q$ *candidates of* $\varepsilon_E$. *Then, we have*

$$\mathbb{P}_{\mathcal{D}}\{e = 0\} \leq U_{SSL}^{OPT} := \min_{\varepsilon_E \in \mathcal{Q}} U_{SSL} \tag{6}$$

*with probability at least* $1 - \delta'_E/Q - \delta'_S/Q$, *where the probability is taken over* $\mathbf{Z}$.

Note that for semi-supervised learning, the upper bound of (B), (C), (D), and (E) in (2) should be provided. The upper bound of (E) is provided in (5), which we denote by $U_{\text{SSL}}$. The upper bound of (B), (C), and (D) are denoted by $w_{\text{SL}}, U_{\text{SL}}$, and $w_{\text{SSL}}$, respectively, each of which is computed by the binomial tail bound. See Algorithm 4 and the proof of Theorem 1 for details.

### 4.4 Neuro-selection Functions

The FDR-E bounds for both supervised and semi-supervised learning are crucial for controlling the final FDR-E of a selective generator given a selection function $\hat{s}$. But, the choice of the selection function is critical for a good selection efficiency and here we discuss a better selection function than the standard one, *i.e.*, $\hat{s}(\mathbf{x}) := \mathbb{1}(f_M(\mathbf{x}, G(\mathbf{x})) \geq \tau_S)$ for $\tau_S \in \mathbb{R}_{\geq 0}$. In particular, certified selective classification [9] considers the single-threshold indicator function using the maximum likelihood as the confidence rate function. For the language generation, the conditional probability of the answer $\hat{\mathbf{y}}$, *i.e.*, $f_{M_1}(\mathbf{x}, \hat{\mathbf{y}})$, would be a natural and commonly-used candidate. However, as it is known to be poorly calibrated [37], an alternative would be a self-consistency

score, *i.e.*, $f_{M_2}(\mathbf{x}, G(\mathbf{x})) \coloneqq \frac{1}{K} \sum_{k=1}^{K} f_E(\tilde{\mathbf{y}}_k, G(\mathbf{x}))$, where $\tilde{\mathbf{y}}_k$ are generated answers with the same question $\mathbf{x}$ but different random seeds. It is empirically shown that the self-consistency score properly quantifies uncertainty when a language model is uncertain of an answer [19]. The importance of score calibration with respect to the true entailment relation is demonstrated in Lemma 4, which provides the sufficient condition for the selective generation algorithm using the single-threshold indicator function (Algorithm 5) to control the FDR-E at *any* level. See Appendix J for a proof.

**Lemma 4.** *If we have access to $E_{true}$ and $f_M$ is perfectly calibrated with respect to $E_{true}$, the FDR-E is monotonically non-increasing in $\tau_S$.*

However, as [37] points out, calibrating the language scoring function remains an uneasy task, os it is still an active research area. Therefore, we propose a general class of selection functions, *neuro-selection functions*, which is the multiple-threshold indicator function using possibly learnable feature map $\Phi : \mathbf{x} \mapsto \mathbb{R}^v$ as follows: $\hat{s}(\mathbf{x}; \Phi, \mathbf{W}, \mathbf{b}) \coloneqq \wedge_{i=1}^{u}(\mathbf{W}\Phi(\mathbf{x}))_i + \mathbf{b}_i \geq 0$, where $\mathbf{W} \in \mathbb{R}^{u \times v}$ and $\mathbf{b} \in \mathbb{R}^{u \times 1}$ are linear proejction and bias terms, respectively. In this paper, we only consider two specific sub-classes of neuro-selection functions, where the former reduces to learning the single-threshold selection function using a scoring function (Algorithm 5) and the latter reduces to learning the bi-threshold selection function using two scoring functions (Algorithm 6). Only the bias term $\mathbf{b}$ is the learnable parameter for both algorithms, where the others set as hyperparameters. Specifically, $\mathbf{W} = \mathbf{I}_1$, $\Phi_1(\mathbf{x}) = [f_M(\mathbf{x}, G(\mathbf{x}))]$, and $\mathbf{b} = -\tau_S$ for Algorithm 5, while $\mathbf{W} = \mathbf{I}_2$, $\Phi_2(\mathbf{x}) = [f_{M_1}(\mathbf{x}, G(\mathbf{x})) \ f_{M_2}(\mathbf{x}, G(\mathbf{x}))]^T$, and $\mathbf{b} = -[\tau_{S,1}, \tau_{S,2}]^T$ for Algorithm 6 if two promising scoring functions exist. Here, developing a selection function learning algorithm where $\mathbf{W}$ and $\Phi(\cdot)$ are also fully learning parameters is left as future work. In the following section, we introduce our algorithm that chooses the optimal combination of scoring functions via neuro-selection functions.

### 4.5 Semi-Supervised Selective Generator Learning Algorithm with Neuro-Selection

$\texttt{SGen}^{\texttt{Semi}}$ is a semi-supervised learning algorithm for certified selective generation, which fully exploits unlabeled data in learning a selection function via certified pseudo-labeling and uses a neuro-selection function for choosing an optimal combination of scoring functions. In particular, $\texttt{SGen}^{\texttt{Semi}}$ solves the following optimization problem over selective generators $\mathcal{H}$ such that $\hat{S}$ closely satisfies the equality in the constraint, as described in Algorithm 7:

$$\mathcal{A}_{\texttt{SGen}^{\texttt{Semi}}} : \quad \text{find}_{\hat{S} \in \mathcal{H}} \ \hat{S} \quad \text{subj. to} \quad w_{\text{SL}} U_{\text{SL}} + w_{\text{SSL}} U_{\text{SSL}}^{\text{OPT}} \leq \varepsilon_S, \tag{7}$$

Here, $\hat{S} \in \mathcal{H}$ has a selection function $\hat{s}(\mathbf{x}; \Phi_2(\mathbf{x}), \texttt{diag}(\mathbf{w}), \mathbf{b})$, where $\mathbf{w} \in \{[1, 0]^T, [0, 1]^T, [1, 1]^T\}$ and $\mathbf{b} \in \mathbb{R}^2_{\leq 0}$. Note that $\texttt{SGen}^{\texttt{Semi}}$ returns an additional term $\hat{U}$, which is the FDR-E bound given the selective generator $\hat{S}$ (*i.e.*, Algorithm 4) and informs the infeasibility of the optimization. The proposed Algorithm 7 satisfies the following controllability guarantee. See Appendix I for a proof.

**Theorem 1.** *$\mathcal{A}_{\texttt{SGen}^{\texttt{Semi}}}$ satisfies the following controllable guarantee on the* FDR-E, *i.e.,*

$$\mathbb{P}\left\{ \mathbb{P}\{G(\mathbf{x}) \notin E_{\text{true}}(\mathbf{y}) \mid \hat{S}(\mathbf{x}) \neq \texttt{IDK}\} \leq \hat{U} \right\} \geq 1 - \delta, \tag{8}$$

where the inner and outer probabilities are taken over $(\mathbf{x}, \mathbf{y}, e, v) \sim \mathcal{D}$ and $\mathbf{Z} \sim \mathcal{D}^n$, respectively, and $(\hat{S}, \hat{U}) \coloneqq \mathcal{A}_{\texttt{SGen}^{\texttt{Semi}}}(\mathbf{Z})$. Here, $\delta \coloneqq \delta_W + \delta_S + \delta_E$ is a desired confidence level, where $\delta_W$ is for the upper bounds on $w_{\text{SL}}$ and $w_{\text{SSL}}$, $\delta_S$ is for (C) in (2) and the NER, and $\delta_E$ is for the FER and FNER.

Here, $\mathcal{A}_{\texttt{SGen}^{\texttt{Semi}}}$ is *controllable* in the sense that it upper-bounds the FDR-E of a learned selective generator to a desired level $\varepsilon_S$ or at least to a minimum achievable level $\hat{U}$ with confidence $\delta$.

## 5 Experiments

We demonstrate the efficacy of our methods in controlling the FDR-E on pre-trained GLMs under various setups. We use two GLMs, GPT-3.5-Turbo and Alpaca-7B, alongside the Natural Questions (NQ) dataset to annotate entailment labels for question-answer pairs. Details on model configurations, datasets, and additional experimental results can be found in Section A.3 and Appendix K.

**Methods.** We consider two heuristic semi-supervised algorithms, $\texttt{SGen}_{\text{PL}}^{\text{H-Semi}}$ and $\texttt{SGen}_{\text{PFL}}^{\text{H-Semi}}$ (Algorithm 9) and an unsupervised learning algorithm [9] $\texttt{SGen}_{\text{EM}}$ (Algorithm 10) as baselines to show the efficacy of our certified semi-supervised method $\texttt{SGen}^{\texttt{Semi}}$ (Algorithm 7). $\texttt{SGen}_{\text{PL}}^{\text{H-Semi}}$ and $\texttt{SGen}_{\text{PFL}}^{\text{H-Semi}}$ exploit the unlabeled data by pseudo-labeling textual entailment based on a threshold as a hyperparameter without any guarantee on mislabeling error. $\texttt{SGen}_{\text{PFL}}^{\text{H-Semi}}$ additionally filters out

Table 1: Comparison results of semi-supervised methods. Here, $|\mathbf{Z}_U| = 10K$ for GPT-3.5-turbo and Alpaca-7B. The best results are highlighted in **bold** and results from methods that do not satisfy desired FDR-E guarantees in learning are underlined.

| Models | | GPT-3.5-turbo | | | | | Alpaca-7B | | | | |
|---|---|---|---|---|---|---|---|---|---|---|---|
| Methods | | Heuristic | | Certified | | | Heuristic | | Certified | | |
| | | $\text{SGen}_{\text{PL}}^{\text{H-Semi}}$ | $\text{SGen}_{\text{PFL}}^{\text{H-Semi}}$ | $\text{SGen}_{\text{EM}}$ | $\text{SGen}_{\text{NoMS}}^{\text{Semi}}$ | $\text{SGen}^{\text{Semi}}$ | $\text{SGen}_{\text{PL}}^{\text{H-Semi}}$ | $\text{SGen}_{\text{PFL}}^{\text{H-Semi}}$ | $\text{SGen}_{\text{EM}}$ | $\text{SGen}_{\text{NoMS}}^{\text{Semi}}$ | $\text{SGen}^{\text{Semi}}$ |
| $f_{M_1}$ | FDR-E | 0.0958 | 0.0283 | 0.1338 | 0.0609 | 0.1589 | 0.0231 | 0.0068 | 0.0359 | 0.0359 | 0.0685 |
| | efficiency | 0.4189 | 0.1719 | 0.5495 | 0.2829 | 0.7334 | 0.0915 | 0.0332 | 0.1580 | 0.1580 | 0.3173 |
| $f_{M_2}$ | FDR-E | 0.1839 | 0.2002 | 0.0914 | 0.1785 | 0.1589 | 0.0698 | 0.0732 | 0.0549 | 0.0698 | 0.0685 |
| | efficiency | 0.7911 | 0.8183 | 0.5332 | 0.7769 | 0.7334 | 0.3207 | 0.3390 | 0.2563 | 0.3200 | 0.3173 |
| average efficiency | | 0.6050 | 0.4951 | — | — | **0.7334** | 0.2061 | 0.1861 | — | — | **0.3173** |

Table 2: Qualitative results by Alpaca7B.

| | | |
|---|---|---|
| **Question x** | Who is the actor who plays Draco Malfoy? | When did the movie Benjamin Button come out? |
| **Correct Answer y** | Thomas Andrew Felton plays Draco Malfoy in the Harry Potter movies. | The movie Benjamin Button come out December 25, 2008 |
| Generated Answer $G(\mathbf{x})$ | The actor who plays Draco Malfoy is Tom Felton. (correct) | The movie The Curious Journey of Benjamin Button was released in 2008. (correct) |
| $\text{SGen}_{\text{EM}}$ [9] | rejected | rejected |
| $\text{SGen}^{\text{Semi}}$ (ours) | accepted | accepted |

a pseudo-labeled sample if its entailment score is below a specific threshold. $\text{SGen}_{\text{EM}}$ is a certified unsupervised method that takes the EM metric for measuring the correctness. We also report results on $\text{SGen}_{\text{NoMS}}^{\text{Semi}}$ (Algorithm 5) for two different scoring functions, $f_{M_1}$ and $f_{M_2}$, used in $\text{SGen}^{\text{Semi}}$. $\text{SGen}_{\text{NoMS}}^{\text{Semi}}$ is a certified semi-supervised learning algorithm using a single-threshold indicator function given a scoring function. We also take $\text{SGen}^{\text{Sup}}$ (Algorithm 8) as a baseline, since it is a direct modification of [9] to the language generation problem.

**Scoring Functions.** We use the conditional probability of an answer as $f_{M_1}$ and the self-consistency score [19] as $f_{M_2}$, since our goal is to generate the sequence which is not only logically consistent to the true answer but also linguistically correct.

**Control Parameters.** To control an FDR-E, we use two user-specified parameters $(\varepsilon, \delta)$, where we use $(0.25, 0.02)$ unless specified. For our methods (*i.e.,* $\text{SGen}^{\text{Semi}}$, $\text{SGen}_{\text{NoMS}}^{\text{Semi}}$, and $\text{SGen}_{\text{NoMS}}^{\text{Semi-Sup}}$), we have five control parameters $(\varepsilon_S, \delta_S, \delta_E, \delta_W)$, where we maps as follows: $\varepsilon_S = \varepsilon$, $\delta_S = (\delta - \delta_W)/2$, $\delta_E = (\delta - \delta_W)/2$, $\delta_W = 10^{-5}$. For other methods without using entailment sets, Algorithm 8, Algorithm 9, and Algorithm 10, we use $\varepsilon$ and $\delta$ accordingly. Additionally, we use $Q = 5$ for Algorithm 3.

**FDR-E Guarantee and Efficiency.** As can be seen in Table 1, our method $\text{SGen}^{\text{Semi}}$ can overall achieve desired FDR-E guarantees with better efficiency compared to baselines. Depending on the quality of scoring functions (*e.g.,* $f_{M_1}$), our variation $\text{SGen}_{\text{NoMS}}^{\text{Semi}}$ may not find a selective generator that satisfies a desired FDR-E (denoted in the underlined FDR-E). The heuristic methods, *i.e.,* $\text{SGen}_{\text{PL}}^{\text{H-Semi}}$ and $\text{SGen}_{\text{PFL}}^{\text{H-Semi}}$, do not provide theoretical guarantees on FDR-E. In Figure 1 and Table 2, we can correctly predict even with the complicated answers, e.g., which have many equivalent words, because we do not rely on the EM metric. We conducted 100 random experiments for each method to show how well FDR-E is bounded under a desired FDR-E. As shown by the green boxes In Figure 4, which are successfully bounded under $\varepsilon_S = 0.25$, we can see that the FDR-E for a learned selective generator is well controlled below $\varepsilon_S$ under the test environment. Among the certified methods with theoretical guarantees, results appear to align well with the expected theoretical basis.

**Why Entailment Labels.** As expected and can be seen in Table 3 by comparing $\text{SGen}_{\text{EM}}$ and $\text{SGen}^{\text{Sup}}$, a metric like EM cannot measure correctness correctly. Unlike classification, generative tasks can have infinite number of true answers so it is not likely to have exact match. Instead, entailment labels provide semantic correctness, so $\text{SGen}^{\text{Sup}}$ can perform better and more efficient than $\text{SGen}_{\text{EM}}$.

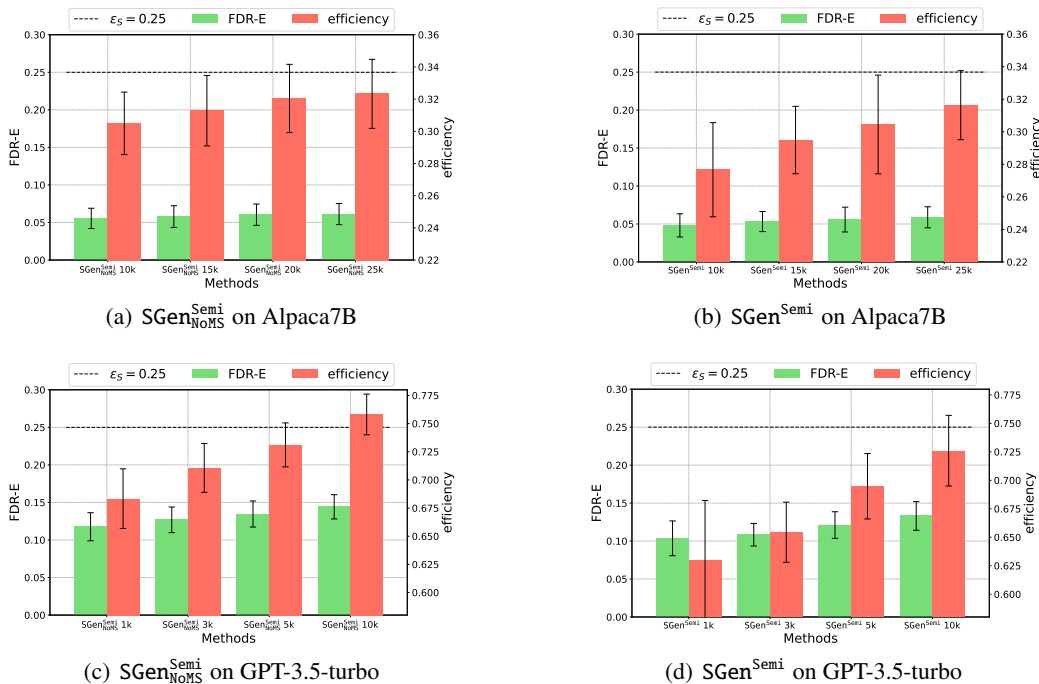

(a) $\texttt{SGen}^{\texttt{Semi}}_{\texttt{NoMS}}$ on Alpaca7B

(b) $\texttt{SGen}^{\texttt{Semi}}$ on Alpaca7B

(c) $\texttt{SGen}^{\texttt{Semi}}_{\texttt{NoMS}}$ on GPT-3.5-turbo

(d) $\texttt{SGen}^{\texttt{Semi}}$ on GPT-3.5-turbo

Figure 3: Efficiency results over different numbers of unlabeled samples. (a) and (b) use $\texttt{SGen}^{\texttt{Semi}}_{\texttt{NoMS}}$ with $f_{M_2}$ score. (c) and (d) use $\texttt{SGen}^{\texttt{Semi}}$ that has neuro-selection function. Both methods show increasing performance as more unlabeled samples $\mathbf{Z}_U$ are used. For each experiment, the values were measured after averaging 10 random splits and an error bar means standard deviation.

**Why Semi-Supervised Learning.** We observe that our semi-supervised learning for selective generation is effective. In particular, the fully supervised methods in Table 3 achieves the efficiency of $0.7535$ and $0.2959$ for GPT-3.5 and Alpaca-7B, respectively, with the entire labeled samples $\mathbf{Z}_E$ (when they satisfy a $\varepsilon$-FDR-E guarantee). Compared to these, the proposed semi-supervised method $\texttt{SGen}^{\texttt{Semi}}$ Table 1 achieves the efficiency of $0.7334$ and $0.3173$ for GPT-3.5 and Alpaca-7B, respectively, by only using $75\%$ of labeled examples. Additionally, we observe that more unlabeled samples are beneficial to achieving better efficiency as can be seen in Figure 3. This implies that if we can approximate the entailment set well and the size of $\mathbf{Z}_U$ is enough, we can enjoy our certified pseudo-entailment labeling by the semi-supervised learning even with small $\mathbf{Z}_E$.

**Why Neuro-Selection.** It is hard to manually find a well calibrated scoring function. But, given multiple scoring functions, a neuro-selection function learns to choose right scoring functions that achieves a desired FDR-E and maximizes selection efficiency. This is empiricially validated in Table 1, as $\texttt{SGen}^{\texttt{Semi}}$ is better on average efficiency.

# 6 Conclusion

We propose selective generation, a generalized version of [9] for GLMs to handle semantic correctness between two structured answers. To this end, we leverage logical entailment to define a new entailment-based FDR (FDR-E) metric. As obtaining entailment labels are expensive, we propose novel semi-supervised learning for selective generation by using entailment sets as a pseudo-labeling function. To enhance the low selective efficiency due to inefficient scoring functions, we propose neuro-selection functions for effectively optimizing scoring functions for better selective efficiency and the FDR-E guarantee. The efficacy of our proposed algorithms $\texttt{SGen}^{\texttt{Semi}}$ and $\texttt{SGen}^{\texttt{Sup}}$ are theoretically and empirically justified.

**Limitations.** Our algorithm needs the i.i.d. assumption for a correctness guarantee, which can be violated in practical situations. We leverage expensive entailment labels, where the labels are obtained by considering logical entailment between a true answer and a generated answer. This limitation is partially mitigated by proposing the semi-supervised method to propagate entailment-labeled samples to samples without entailment labels. Also, our results show the empirical FDR-E is not much closely bounded under $\varepsilon$, especially on Alpaca7B, which implies that we may need a tighter FDR-E bound.

## Acknowledgements

This work was supported by Institute of Information & communications Technology Planning & Evaluation (IITP) grant funded by the Korea government (MSIT) (No.RS-2019-II191906, Artificial Intelligence Graduate School Program (POSTECH) (50%); RS-2024-00457882, National AI Research Lab Project (25%); RS-2024-00509258, Global AI Frontier Lab (25%)). Also, we appreciate valuable comments by NeurIPS reviewers.

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

# A  Discussion

## A.1  Conformal Prediction

Conformal prediction [10] provides a promising way to quantify uncertainty of a model with a correctness guarantee under minimal assumptions. Here, we consider PAC prediction sets [11], an interpretation of tolerance region [31] and training-conditional inductive conformal prediction [20] in the lens of PAC learning theory [32] (*i.e.,* learning a "good" function within a function family from data). This interpretation inspires us to generalize selective generation for GLMs via neural selection functions.

**Conformal Set Model.** We consider a *conformal (prediction) set model* $\hat{C} : \mathcal{X} \to 2^{\mathcal{Y}}$ that measures the uncertainty of a target model; in conformal prediction, this model is specifically called a *scoring function* $f : \mathcal{X} \times \mathcal{Y} \to \mathbb{R}_{\geq 0}$ that measures the conformity (or likelihood) of $\mathbf{x}$ for being $\mathbf{y}$ with respect to $f$; thus, $f(\mathbf{x}, \mathbf{y})$ is called a *conformity score*. In particular, we consider scalar parameterization of a conformal set [11, 33] as follows: $C(\mathbf{x}) \coloneqq \{\mathbf{y} \in \mathcal{Y} \mid f(\mathbf{x}, \mathbf{y}) \geq \tau\}$, where $\tau \in \mathbb{R}_{\geq 0}$ is a scalar parameter.

**Conformal Sets and Uncertainty.** The output of the conformal set model is a set of labels, which naturally represents the *uncertainty of a scoring function on an example* via the size of a conformal set. In particular, if the scoring function $f$ is unsure on its prediction on $\mathbf{x}$ (due to uncertainty on a label distribution of $\mathbf{x}$, *i.e.,* aleatoric uncertainty, and due to uncertainty in the modeling of $f$, *i.e.,* epistemic uncertainty), the conformal set is larger than it is when the scoring function is sure on its prediction.

To be precise, we consider a *true conformal set* $C^*(\mathbf{x}) \coloneqq \{\mathbf{y} \in \mathcal{Y} \mid f(\mathbf{x}, \mathbf{y}) \geq f(\mathbf{x}, \mathbf{y}^*)\}$, where $\mathbf{y}^*$ is the true label of $x$. In particular, the true conformal set is a minimal set that contains a true label and labels with larger scores than the true label score; thus, the size of the true conformal set intuitively measures the uncertainty of a scoring function on the given example, *i.e.,* the scoring function's possibilities on making wrong predictions, instead of the true prediction.

The true conformal set clearly captures the uncertainty, but the true label is unknown in inference time. Thus, the true conformal set is approximated via scalar parameterization [11, 33] as follows:

$$C(\mathbf{x}) \coloneqq \{\mathbf{y} \in \mathcal{Y} \mid f(\mathbf{x}, \mathbf{y}) \geq \tau\}, \tag{9}$$

where $\tau \in \mathbb{R}_{\geq 0}$ is a scalar parameter.

**Correctness.** As we desire to construct a conformal set close to the true conformal set, we define the correctness of the conformal set based on its similarity to the true one. In particular, we wish to have the smallest $C(\mathbf{x})$ such that $C^*(\mathbf{x}) \subseteq C(\mathbf{x})$, or equivalently $C(\mathbf{x})$ needs to have the smallest $\tau$ while $y \in C(\mathbf{x})$. This correctness definition is realized into two ways: a coverage guarantee [10] or a PAC guarantee [20].

**Assumption.** We assume that samples are independent and identically distributed (i.i.d.), *i.e.,* the i.i.d. assumption. In particular, all samples for testing and learning prediction sets are independently drawn from the same but known distribution $\mathcal{D}$.

**PAC guarantee.** Under the i.i.d. assumption, we learn a conformal set $\hat{C}$ that includes the most true labels (*approximately correct*). In particular, this means that the miscoverage of $\hat{C}$ is less than a desired level $\varepsilon \in (0, 1)$, *i.e.,* $\mathcal{R}_{\mathrm{MC}}(\hat{C}) \coloneqq \mathbb{P}\{\mathbf{y} \notin \hat{C}(\mathbf{x})\} \leq \varepsilon$, where the probability is taken over i.i.d. samples $(\mathbf{x}, \mathbf{y}) \sim \mathcal{D}$. This risk on micoverage can be generalized to be the risk on indicator loss, $\mathcal{R}_{01}(\hat{C}) \coloneqq \mathbb{E}_{\mathcal{D}} \ell_{01}(\hat{C}, \mathbf{x}, \mathbf{y})$. Here, the conformal set $\hat{C}$ is learned from a randomly drawn calibration set, so we desire to construct $\hat{C}$ that has a desired error for the most of random calibration sets (*probably approximately correct*), *i.e.,* $\mathbb{P}\{\mathcal{R}_{01}(\hat{C}) \leq \varepsilon\} \geq 1 - \delta$, where $\delta \in (0, 1)$ is a desired confidence level and the probability is taken over $n$ i.i.d. calibration samples $\mathbf{Z} \sim \mathcal{D}^n$, used to learn $\hat{C}$.

**Algorithm.** The PAC conformal prediction set method [11, 34] considers the following algorithm $\mathcal{A}_{\mathrm{Binom}} : (\mathcal{X} \times \mathcal{Y})^* \to \mathcal{H}$ to learn a conformal set model $\hat{C}$, parameterized by $\hat{\tau}$, where $\mathcal{H}$ is a finely-discretized $\mathbb{R}_{\geq 0}$:

$$\mathcal{A}_{\mathrm{Binom}}{}^1 : \quad \hat{\tau} = \max_{\tau \in \mathcal{H}} \tau \quad \text{subj. to} \quad U_{\mathrm{Binom}}(k_\tau; n, \delta) \leq \varepsilon, \tag{10}$$

---

[1] $\mathcal{A}_{\mathrm{Binom}}$ returns $\hat{\tau} = 0$ if it is infeasible.

where $k_\tau := \sum_{i=1}^n \ell_{01}(\hat{C}, \mathbf{x}_i, \mathbf{y}_i)$. Here, $U_{\text{Binom}}$ is a binomial tail bound, *i.e.,* $\mathbb{P}\{\mathcal{R}_{01}(C) \leq U_{\text{Binom}}(k_\tau; n, \delta)\} \geq 1 - \delta$ for any $C$, where $U_{\text{Binom}}(k; n, \delta) :=$ $\inf\{\theta \in [0,1] \,|\, F(k; n, \theta) \leq \delta\} \cup \{1\}$ and $F(k; n, \theta)$ is a cumulative distribution function (CDF) of a binomial distribution with $n$ trials and success probability $\theta$. This algorithm is PAC.

**Theorem 2.** *([11, 20, 34]) The algorithm $\mathcal{A}_{Binom}$ is PAC, i.e., for any $f$, $\varepsilon \in (0,1)$, $\delta \in (0,1)$, and $n \in \mathbb{Z}_{\geq 0}$, we have $\mathbb{P}\{\mathcal{R}_{01}(\hat{C}) \leq \varepsilon\} \geq 1 - \delta$, where the probability is taken over i.i.d. labeled examples $\mathbf{Z} \sim \mathcal{D}^n$, and $\hat{C} = \mathcal{A}_{Binom}(\mathbf{Z})$.*

Here, we slightly generalize the known PAC guarantee to hold for any risk with indicator loss. See Appendix F for a proof. Note that the PAC guarantee generally holds only if an enough number of samples is provided (when we know a function family including a true function). However, we consider PAC algorithms that hold for any number of samples due to the structural property of prediction sets, *i.e.,* a prediction set is always correct if $\tau = 0$ (thus $\hat{C}(\mathbf{x}) = \mathcal{Y}$), regardless of the sample size. In other words, if the calibration samples are not sufficient, the prediction set is constructed to return $\mathcal{Y}$ to satisfy the PAC guarantee.

## A.2 Sample Space Decomposition

Given the generator $G$ and the entailment set function $\hat{E}$, the sample space $\Omega := \mathcal{X} \times \mathcal{Y} \times \mathcal{E} \times \mathcal{V}$ can be partitioned as follows:

$$
\begin{aligned}
\Omega &= \underbrace{\{(\mathbf{x}, \mathbf{y}, e, v) \mid G(\mathbf{x}) \in E_{\text{true}}(\mathbf{y})\}}_{\Omega_{\text{TD}}^{E_{\text{true}}}} \cup \underbrace{\{(\mathbf{x}, \mathbf{y}, e, v) \mid G(\mathbf{x}) \notin E_{\text{true}}(\mathbf{y})\}}_{\Omega_{\text{FD}}^{E_{\text{true}}}} \\
&= \underbrace{\{(\mathbf{x}, \mathbf{y}, e, v) \mid e = 0\}}_{\Omega_{\text{TD}}^{E_{\text{true}}}} \cup \underbrace{\{(\mathbf{x}, \mathbf{y}, e, v) \mid e = 1\}}_{\Omega_{\text{FD}}^{E_{\text{true}}}} \\
&= \underbrace{\underbrace{\{(\mathbf{x}, \mathbf{y}, e, v) \mid e = 1 \text{ and } G(\mathbf{x}) \in \hat{E}(\mathbf{y})\}}_{\Omega_{\text{TE}}^{\hat{E}}} \cup \underbrace{\{(\mathbf{x}, \mathbf{y}, e, v) \mid e = 1 \text{ and } G(\mathbf{x}) \notin \hat{E}(\mathbf{y})\}}_{\Omega_{\text{FNE}}^{\hat{E}}}}_{\Omega_{\text{TD}}} \cup \\
&\quad \underbrace{\underbrace{\{(\mathbf{x}, \mathbf{y}, e, v) \mid e = 0 \text{ and } G(\mathbf{x}) \notin \hat{E}(\mathbf{y})\}}_{\Omega_{\text{TNE}}^{\hat{E}}} \cup \underbrace{\{(\mathbf{x}, \mathbf{y}, e, v) \mid e = 0 \text{ and } G(\mathbf{x}) \in \hat{E}(\mathbf{y})\}}_{\Omega_{\text{FE}}^{\hat{E}}}}_{\Omega_{\text{FD}}} \\
&= \underbrace{\{\Omega_{\text{TE}}^{\hat{E}} \cup \Omega_{\text{FE}}^{\hat{E}}\}}_{\Omega_{\text{TD}}^{\hat{E}}} \cup \underbrace{\{\Omega_{\text{FNE}}^{\hat{E}} \cup \Omega_{\text{TNE}}^{\hat{E}}\}}_{\Omega_{\text{FD}}^{\hat{E}}}.
\end{aligned}
$$

Here, the short-hands are defined as follows:

- True discovery rate (TDR): $\mathbb{P}(\Omega_{\text{TD}}^{E_{\text{true}}})$
- False discovery rate (FDR): $\mathbb{P}(\Omega_{\text{FD}}^{E_{\text{true}}})$
- True entailment rate (TER): $\mathbb{P}(\Omega_{\text{TE}}^{\hat{E}})$
- False non-entailment rate (FNER): $\mathbb{P}(\Omega_{\text{FNE}}^{\hat{E}})$
- True non-entailment rate (TNER): $\mathbb{P}(\Omega_{\text{TNE}}^{\hat{E}})$
- False entailment rate (FER): $\mathbb{P}(\Omega_{\text{FER}}^{\hat{E}})$

## A.3 Experiment Setup

### A.3.1 Computing Environment

Our system environment consists of 4 NVIDIA A100 80GB with 128 CPUs.

### A.3.2 Models and Datasets

We use two large language models (LLMs), *GPT-3.5-Turbo* and *Alpaca-7B*, for language generation. We use deberta-v2-xxlarge-mnli as our entailment model.

For each GLM to annotate entailment labels for each question, answer, and generated answer pair, we annotate entailment labels. Specifically, we consider the open-ended QA task, where the model is prompted to generate the answer in a declarative form given a question. To validate our method and its theoretical guarantee on controlling FDR-E, we create a dataset on textual entailment using the Natural Questions (NQ) dataset [17] for each GLM. Based on the transformation method by [38] that converts the question and answer pair in QA dataset into a declarative form, we manually labeled textual entailment by letting the generated sequence as the premise and the reference answer in declarative form as the hypothesis. Similar work can be found in [39], but they label the textual entailment based on the extractive answer from the model. Approximately 7.3k (7,374) and 4.6k (4,595) samples are labeled for *Alpaca-7B* and *GPT-3.5-Turbo*, respectively, and both are split into calibration and test data at an 8:2 ratio. For semi-supervised learning algorithms that exploit unlabeled data (Algorithm 7, Algorithm 9), at most 27k and 10k unlabeled samples are used to train a selective generator, varying its size. Besides, semi-supervised learning algorithms use only 75% of the labeled calibration data compared to what is used by supervised methods (Algorithm 8, Algorithm 10).

# B   Semi-supervised Selective Generation Algorithms (Certified)

---

**Algorithm 1** Entailment Set Learning with a False Entailment Rate (FER) Guarantee

---

1: **procedure** ES($f_E, \mathbf{Z}_E, \varepsilon_E, \delta_E$)
2: $\quad \mathbf{Z}_E \leftarrow \text{SORT}_{f_E}(\mathbf{Z}_E)$ $\qquad\qquad\qquad\qquad$ ($\triangleright$) In an increasing order of $f_E(\mathbf{y}_i, G(\mathbf{x}_i))$
3: $\quad (\underline{i}, \bar{i}) \leftarrow (1, |\mathbf{Z}_E|)$
4: $\quad$ **for** $i = 1$ **to** $\lceil \log|\mathbf{Z}_E| \rceil$ **do**
5: $\qquad k^{(i)} \leftarrow \sum_{(\mathbf{x},\mathbf{y},e) \in \mathbf{Z}_E} \mathbb{1}(e = 0, f_E(G(\mathbf{x}), \mathbf{y}) \geq f_E(G(\mathbf{x}_{\lceil (\underline{i}+\bar{i})/2 \rceil}), \mathbf{y}_{\lceil (\underline{i}+\bar{i})/2 \rceil}))$
6: $\qquad U \leftarrow U_{\text{Binom}}(k^{(i)}, |\mathbf{Z}_E|, \delta_E)$
7: $\qquad$ **if** $U \leq \varepsilon_E$ **then**
8: $\qquad\qquad \bar{i} \leftarrow \lceil (\underline{i} + \bar{i})/2 \rceil$
9: $\qquad$ **else**
10: $\qquad\qquad \underline{i} \leftarrow \lceil (\underline{i} + \bar{i})/2 \rceil$
11: $\quad$ **return** $\tau_E$

---

**Algorithm 2** $U_{\text{SSL}}$ Computation (for Single $\varepsilon_E$)

---

1: **procedure** COMPUTE-$U_{\text{SSL}}$($f_E, \mathbf{Z}_E, \mathbf{Z}_U, \delta_S, \varepsilon_E, \delta_E$)
2: $\quad \tau_E \leftarrow \text{ES}(f_E, \mathbf{Z}_E, \varepsilon_E, \delta_E/2)$
3: $\quad \ell \leftarrow \sum_{(\mathbf{x},\mathbf{y},e) \in \mathbf{Z}_E} \mathbb{1}(e = 1, f_E(G(\mathbf{x}), \mathbf{y}) < \tau_E)$
4: $\quad k \leftarrow \sum_{(\mathbf{x},\mathbf{y}) \in \mathbf{Z}_U} \mathbb{1}(f_E(G(\mathbf{x}), \mathbf{y}) < \tau_E)$
5: $\quad U_{\text{SSL}} \leftarrow \varepsilon_E - L_{\text{Binom}}(\ell; |\mathbf{Z}_E|, \delta_E/2) + U_{\text{Binom}}(k, |\mathbf{Z}_U|, \delta_S/2)$
6: $\quad$ **return** $U_{\text{SSL}}$

---

**Algorithm 3** Optimal $U_{\text{SSL}}$ Search

---

1: **procedure** COMPUTE-$U_{\text{SSL}}^{\text{OPT}}$($f_E, \mathbf{Z}_E, \mathbf{Z}_U, \delta_S, Q, \delta_E$)
2: $\quad \mathbf{Z}_E \leftarrow \text{SORT}_{f_E}(\mathbf{Z}_E)$ $\qquad\qquad\qquad\qquad$ ($\triangleright$) In an increasing order of $f_E(\mathbf{y}_i, G(\mathbf{x}_i))$
3: $\quad (\underline{i}, \bar{i}) \leftarrow (1, |\mathbf{Z}_E|)$
4: $\quad \varepsilon_{\max} \leftarrow \sum_{(\mathbf{x},\mathbf{y},e) \in \mathbf{Z}_E} \mathbb{1}(e = 0)/|\mathbf{Z}_E|$
5: $\quad \mathcal{H}_E \leftarrow \{\varepsilon_1 = \varepsilon_{\max}, \dots, \varepsilon_Q = 1/|Q|\varepsilon_{\max}\}$
6: $\quad U_{\text{SSL}}^{\text{OPT}} \leftarrow \infty$
7: $\quad$ **for** $i$ **in** $\{1, \dots, Q\}$ **do**
8: $\qquad U_{\text{SSL}}^{(i)} \leftarrow \text{Compute-}U_{\text{SSL}}(f_E, \mathbf{Z}_E, \mathbf{Z}_U, \delta_S/Q, \varepsilon_i, \delta_E/Q)$
9: $\qquad$ **if** $U_{\text{SSL}}^{(i)} \leq U_{\text{SSL}}^{\text{OPT}}$ **then**
10: $\qquad\qquad U_{\text{SSL}}^{\text{OPT}} \leftarrow U_{\text{SSL}}^{(i)}$
11: $\quad$ **return** $U_{\text{SSL}}^{\text{OPT}}$

---

---

**Algorithm 4** FDR-E Bound Computation

---

1: **procedure** FDR-E-BOUND($f_E, \mathbf{Z}_E, \mathbf{Z}_U, \delta_S, Q, \delta_E, \delta_W$)
2:     $w_{\text{SL}} \leftarrow U_{\text{Binom}}(|\mathbf{Z}_E|; |\mathbf{Z}_E| + |\mathbf{Z}_U|, \delta_W/2)$            ($\triangleright$) Upper bound of (B) in (2)
3:     $k_{\text{SL}} \leftarrow \sum_{(\mathbf{x},\mathbf{y},e)\in\mathbf{z}_E} \mathbb{1}(e=0)$
4:     $U_{\text{SL}} \leftarrow U_{\text{Binom}}(k_{\text{SL}}; |\mathbf{Z}_E|, \delta_S/2)$                  ($\triangleright$) Upper bound of (C) in (2)
5:     $w_{\text{SSL}} \leftarrow U_{\text{Binom}}(|\mathbf{Z}_U|; |\mathbf{Z}_E| + |\mathbf{Z}_U|, \delta_W/2)$        ($\triangleright$) Upper bound of (D) in (2)
6:     $U_{\text{SSL}}^{\text{OPT}} \leftarrow \texttt{Compute} - U_{\text{SSL}}^{\text{OPT}}(f_E, \mathbf{Z}_E, \mathbf{Z}_U, \delta_S/2, Q, \delta_E/2)$    ($\triangleright$) Upper bound of (E) in (2)
7:     $U \leftarrow w_{\text{SL}} U_{\text{SL}} + w_{\text{SSL}} U_{\text{SSL}}^{\text{OPT}}$
8:     **return** $U$

---

---

**Algorithm 5** Semi-supervised Selective Generator Learning (Single-threshold Selection Function)

---

1: **procedure** SGEN-SEMI($f_M, f_E, G, \mathbf{Z}_E, \mathbf{Z}_U, \varepsilon_S, \delta_S, Q, \delta_E, \delta_W, \texttt{return\_bool} = \texttt{False}$)
2:     $\mathbf{Z}_{U,E} \leftarrow \mathbf{Z}_U \cup \mathbf{Z}_E$
3:     $\mathbf{Z}_{U,E} \leftarrow \text{SORT}_{f_M}(\mathbf{Z}_{U,E})$             ($\triangleright$) In an increasing order of $f_M(\mathbf{y}_i, G(\mathbf{x}_i))$
4:     $(\underline{i}, \bar{i}) \leftarrow (1, \mathbf{Z}_{U,E})$
5:     $U_{\min} \leftarrow \infty; \tau_{\min} \leftarrow \texttt{NULL}$
6:     **for** $i = 1$ **to** $\lceil \log_2 \mathbf{Z}_{U,E} \rceil$ **do**
7:         $\tau_S^{(i)} \leftarrow f_M(\mathbf{x}_{\lceil(\underline{i}+\bar{i})/2\rceil}, G(\mathbf{x}_{\lceil(\underline{i}+\bar{i})/2\rceil}))$
8:         $\mathbf{Z}_E^{(i)} \leftarrow \{(\mathbf{x}, \mathbf{y}, e) \in \mathbf{Z}_E \mid f_M(\mathbf{x}, G(\mathbf{x})) \geq \tau_S^{(i)}\}$
9:         $\mathbf{Z}_U^{(i)} \leftarrow \{(\mathbf{x}, \mathbf{y}) \in \mathbf{Z}_U \mid f_M(\mathbf{x}, G(\mathbf{x})) \geq \tau_S^{(i)}\}$
10:       $U^{(i)} \leftarrow \text{FDR-E-BOUND}(f_E, \mathbf{Z}_E^{(i)}, \mathbf{Z}_U^{(i)}, \frac{\delta_S}{\lceil\log_2|\mathbf{Z}_{U,E}|\rceil}, Q, \frac{\delta_E}{\lceil\log_2 \mathbf{Z}_{U,E}\rceil}, \frac{\delta_W}{\lceil\log_2 \mathbf{Z}_{U,E}\rceil})$
11:         **if** $U^{(i)} \leq U_{\min}$ **then**
12:             $U_{\min} \leftarrow U^{(i)}; \; \tau_{\min} \leftarrow \tau_S^{(i)}$
13:         **if** $U^{(i)} \leq \varepsilon_S$ **then**
14:             $\bar{i} \leftarrow \lceil(\underline{i}+\bar{i})/2\rceil$
15:         **else**
16:             $\underline{i} \leftarrow \lceil(\underline{i}+\bar{i})/2\rceil$
17:     $\tau_S \leftarrow \tau_S^{(i)}$
18:     **if** $U_{\min} \leq \varepsilon_S$ **then**
19:         $\hat{U} \leftarrow U^{(i)}$
20:         $\texttt{Bounded} \leftarrow \texttt{Success}$
21:     **else**
22:         $\hat{U} \leftarrow U_{\min}$
23:         $\tau_S \leftarrow \tau_{\min}$
24:         $\texttt{Bounded} \leftarrow \texttt{Fail}$
25:     **return** $(\tau_S, \hat{U}, \texttt{Bounded})$ if $\texttt{return\_bool}$ else $(\tau_S, \hat{U})$.

---

**Algorithm 6** Semi-supervised Selective Generator Learning (Double-threshold Selection Function)

1: **procedure** SGEN-SEMI2($f_{M_1}$, $f_{M_2}$, $f_E$, $G$, $\mathbf{Z}_E$, $\mathbf{Z}_U$, $\varepsilon_S$, $\delta_S$, $Q$, $\delta_E$, $\delta_W$, `return_bool =` `False`)

2:     $\mathbf{Z}_{U,E} \leftarrow \mathbf{Z}_U \cup \mathbf{Z}_E$

3:     $\mathbf{Z}_{U_1,E_1} \leftarrow \text{SORT}_{f_{M_1}}(\mathbf{Z}_{U,E})$                          ($\triangleright$) In an increasing order of $f_{M_1}(\mathbf{y}_i, G(\mathbf{x}_i))$

4:     $\mathbf{Z}_{U_2,E_2} \leftarrow \text{SORT}_{f_{M_2}}(\mathbf{Z}_{U,E})$                          ($\triangleright$) In an increasing order of $f_{M_2}(\mathbf{y}_i, G(\mathbf{x}_i))$

5:     $U_{\min} \leftarrow \infty$; $\tau_{\min} \leftarrow \text{NULL}$

6:     $(\underline{i}, \overline{i}) \leftarrow (1, |\mathbf{Z}_{U_1,E_1}|)$

7:     $I \leftarrow \lceil \log_2 |\mathbf{Z}_{U,E}| \rceil$

8:     **for** $i = 1$ **to** $\lceil \log_2 |\mathbf{Z}_{U,E}| \rceil$ **do**

9:         $\tau_S^{(i)} \leftarrow f_{M_1}(\mathbf{x}_{\lceil (\underline{i}+\overline{i})/2 \rceil}, G(\mathbf{x}_{\lceil (\underline{i}+\overline{i})/2 \rceil}))$

10:         $U_{\min}^{(i)} \leftarrow \infty$; $\tau_{\min}^{(i)} \leftarrow \text{NULL}$

11:         $(\underline{j}, \overline{j}) \leftarrow (1, |\mathbf{Z}_{U_2,E_2}|)$

12:         **for** $j = 1$ **to** $\lceil \log_2 |\mathbf{Z}_{U,E}| \rceil$ **do**

13:             $\tau_S^{(j)} \leftarrow f_{M_2}(\mathbf{x}_{\lceil (\underline{j}+\overline{j})/2 \rceil}, G(\mathbf{x}_{\lceil (\underline{j}+\overline{j})/2 \rceil}))$

14:             $\mathbf{Z}_E^{(i,j)} \leftarrow \{(\mathbf{x}, \mathbf{y}, e) \in \mathbf{Z}_E \mid \hat{s}(\mathbf{x}; G, f_{M_1}, f_{M_2}, \tau_S^{(i)}, \tau_S^{(j)}) = 1\}$

15:             $\mathbf{Z}_U^{(i,j)} \leftarrow \{(\mathbf{x}, \mathbf{y}) \in \mathbf{Z}_U \mid \hat{s}(\mathbf{x}; G, f_{M_1}, f_{M_2}, \tau_S^{(i)}, \tau_S^{(j)}) = 1\}$

16:             $U^{(i,j)} \leftarrow \text{FDR-E-BOUND}(f_E, \mathbf{Z}_E^{(i,j)}, \mathbf{Z}_U^{(i,j)}, \frac{\delta_S}{I^2}, Q, \frac{\delta_E}{I^2}, \frac{\delta_W}{I^2})$

17:             **if** $U^{(i,j)} \leq U_{\min}^{(i)}$ **then**

18:                 $U_{\min}^{(i)} \leftarrow U^{(i,j)}$; $\tau_{\min}^{(i)} \leftarrow (\tau_S^{(i)}, \tau_S^{(j)})$

19:             **if** $U^{(i,j)} \leq \varepsilon_S$ **then**

20:                 $\overline{j} \leftarrow \lceil (\underline{j} + \overline{j})/2 \rceil$

21:             **else**

22:                 $\underline{j} \leftarrow \lceil (\underline{j} + \overline{j})/2 \rceil$

23:         **if** $U_{\min}^{(i)} \leq U_{\min}$ **then**

24:             $U_{\min} \leftarrow U_{\min}^{(i)}$; $\tau_{\min} \leftarrow \tau_{\min}^{(i)}$

25:         **if** $i \neq \lceil \log_2 |\mathbf{Z}_{U,E}| \rceil$ **then**

26:             **if** $U_{\min}^{(i)} \leq \varepsilon_S$ **then**

27:                 $\overline{i} \leftarrow \lceil (\underline{i} + \overline{i})/2 \rceil$

28:             **else**

29:                 $\underline{i} \leftarrow \lceil (\underline{i} + \overline{i})/2 \rceil$

30:         **else**

31:             $\tau_S \leftarrow (\tau_S^{(i)}, \tau_S^{(j)})$

32:     **if** $U_{\min} \leq \varepsilon_S$ **then**

33:         $\hat{U} \leftarrow U^{(i,j)}$; `Bounded` $\leftarrow$ `Success`

34:     **else**

35:         $\hat{U} \leftarrow U_{\min}$; $\tau_S \leftarrow \tau_{\min}$; `Bounded` $\leftarrow$ `Fail`

36:     **return** $(\tau_S, \hat{U}, \texttt{Bounded})$ if `return_bool` else $(\tau_S, \hat{U})$

**Algorithm 7** Semi-supervised Selective Generator Learning with Neuro-Selection
___

1: **procedure** SGEN-SEMI-MS($f_{M_1}, f_{M_2}, f_E, G, \mathbf{Z}_E, \mathbf{Z}_U, \varepsilon_S, \delta_S, Q, \delta_E, \delta_W$)
2: $\quad$ $\mathcal{M}_{\text{Success}} = \{\}$; $\mathcal{M}_{\text{Fail}} = \{\}$
3: $\quad$ $(\tau_{S_1}, \hat{U}_1, \text{Bounded}_1) \leftarrow \texttt{SGen-Semi}(f_{M_1}, f_E, G, \mathbf{Z}_E, \mathbf{Z}_U, \varepsilon_S, \delta_S/3, Q, \delta_E/3, \delta_W/3, \texttt{return\_bool} = \texttt{True})$
4: $\quad$ $(\tau_{S_2}, \hat{U}_2, \text{Bounded}_2) \leftarrow \texttt{SGen-Semi}(f_{M_2}, f_E, G, \mathbf{Z}_E, \mathbf{Z}_U, \varepsilon_S, \delta_S/3, Q, \delta_E/3, \delta_W/3, \texttt{return\_bool} = \texttt{True})$
5: $\quad$ $(\tau_{S_3}, \hat{U}_3, \text{Bounded}_3) \leftarrow \texttt{SGen-Semi2}(f_{M_1}, f_{M_2}, f_E, G, \mathbf{Z}_E, \mathbf{Z}_U, \varepsilon_S, \delta_S/3, Q, \delta_E/3, \delta_W/3, \texttt{return\_bool} = \texttt{True})$
6: $\quad$ $\mathcal{M} := \{(\tau_{S_1}, \hat{U}_1, s_1, \text{Bounded}_1), (\tau_{S_2}, \hat{U}_2, s_2, \text{Bounded}_2), (\tau_{S_3}, \hat{U}_3, s_3, \text{Bounded}_3)\}$
7: $\qquad\qquad\qquad\qquad\qquad\qquad\qquad$ $(\triangleright)$ $s_i$ refers to the scoring function(s) used in each algorithm.
8: $\quad$ **for** $(\tau_S, \hat{U}, s, \text{Bounded})$ **in** $\mathcal{M}$ **do**
9: $\qquad$ **if** $\text{Bounded} = \text{Success}$ **then**
10: $\qquad\quad$ $\mathcal{M}_{\text{Success}} \leftarrow \mathcal{M}_{\text{Success}} \cup \{(\tau_S, \hat{U}, s)\}$
11: $\qquad$ **else**
12: $\qquad\quad$ $\mathcal{M}_{\text{Fail}} \leftarrow \mathcal{M}_{\text{Fail}} \cup \{(\tau_S, \hat{U}, s)\}$
13: $\quad$ **if** $\mathcal{M}_{\text{Success}} = \{\}$ **then**
14: $\qquad$ **return** $(\tau_S, \hat{U}, s) \leftarrow \arg\min_{(\tau_S, \hat{U}, s) \in \mathcal{M}_{\text{Fail}}} \hat{U}$
15: $\quad$ **else**
16: $\qquad$ **return** $(\tau_S, \hat{U}, s) \leftarrow \arg\max_{(\tau_S, \hat{U}, s) \in \mathcal{M}_{\text{Success}}} \hat{U}$
___

## C    Supervised Selective Generation Algorithms (Certified)

---

**Algorithm 8** Supervised Selective Generator Learning with $\mathcal{R}_{R_E}(\hat{S})$ Control

---

1: **procedure** SG-SUP$(f_M, G, \mathbf{Z}_E, \varepsilon, \delta)$
2:     $(\underline{i}, \bar{i}) \leftarrow (1, |\mathbf{Z}_E|)$
3:     **for** $i = 1$ **to** $\lceil \log_2 |\mathbf{Z}_E| \rceil$ **do**
4:         $\tau_S^{(i)} \leftarrow f_M(\mathbf{x}_{\lceil (\underline{i}+\bar{i})/2 \rceil}, G(\mathbf{x}_{\lceil (\underline{i}+\bar{i})/2 \rceil}))$
5:         $\mathbf{Z}_E^{(i)} \leftarrow \{(\mathbf{x}, \mathbf{y}, e) \in \mathbf{Z}_E \mid f_M(\mathbf{x}, G(\mathbf{x})) \geq \tau_S^{(i)}\}$
6:         $k^{(i)} \leftarrow \sum_{(\mathbf{x}, \mathbf{y}, e) \in \mathbf{Z}_E} \mathbb{1}(e = 0)$
7:         $U^{(i)} \leftarrow U_{\text{Binom}}(k^{(i)}; |\mathbf{Z}_E^{(i)}|, \delta / \lceil \log_2 |\mathbf{Z}_E| \rceil)$
8:         **if** $U^{(i)} \leq \varepsilon$ **then**
9:             $\bar{i} \leftarrow \lceil (\underline{i}+\bar{i})/2 \rceil$
10:         **else**
11:             $\underline{i} \leftarrow \lceil (\underline{i}+\bar{i})/2 \rceil$
12:     $\tau_S \leftarrow \tau_S^{(i)}$
13:     $\hat{U} \leftarrow U^{(i)}$
14:     **return** $\tau_S, \hat{U}$

---

# D Semi-supervised Selective Generation Algorithms (Heuristic)

---

**Algorithm 9** Semi-supervised Selective Generator Learning with Pseudo-entailment Labels

---

1: **procedure** SG-PSL-H-SEMI($f_M$, $f_E$, $G$, $\mathbf{Z}_E$, $\mathbf{Z}_U$, $\varepsilon$, $\delta$, $\tau_{\text{PL}}$, FILTER)
2:     **if** FILTER $==$ TRUE **then**
3:         $\mathbf{Z}_U \leftarrow \{(\mathbf{x}, \mathbf{y}) \mid f_E(G(\mathbf{x}), \mathbf{y}) \geq \tau_{\text{PL}} \text{ or } 1 - f_E(G(\mathbf{x}), \mathbf{y}) \geq \tau_{\text{PL}}\}$
4:     $\mathbf{Z}_U \leftarrow \{(\mathbf{x}, \mathbf{y}, \tilde{e}) \mid (\mathbf{x}, \mathbf{y}) \in \mathbf{Z}_U, \tilde{e} = \mathbb{1}(f_E(G(\mathbf{x}), \mathbf{y}) \geq \tau_{\text{PL}})\}$
5:     $\mathbf{Z}_E \leftarrow \{(\mathbf{x}, \mathbf{y}, \tilde{e}) \mid (\mathbf{x}, \mathbf{y}, e) \in \mathbf{Z}_U, \tilde{e} = e\}$
6:     $\mathbf{Z}_{U,E} \leftarrow \text{SORT}_{f_M}(\mathbf{Z}_E \cup \mathbf{Z}_U)$
7:     $(\underline{i}, \bar{i}) \leftarrow (1, |\mathbf{Z}_{U,E}|)$
8:     **for** $i = 1$ **to** $\lceil \log_2 |\mathbf{Z}_{U,E}| \rceil$ **do**
9:         $\tau_S^{(i)} \leftarrow f_M(\mathbf{x}_{\lceil (\underline{i}+\bar{i})/2 \rceil}, G(\mathbf{x}_{\lceil (\underline{i}+\bar{i})/2 \rceil}))$
10:        $\mathbf{Z}_{U,E}^{(i)} \leftarrow \{(\mathbf{x}, \mathbf{y}) \in \mathbf{Z}_{U,E} \mid f_M(\mathbf{x}, G(\mathbf{x})) \geq \tau_S^{(i)}\}$
11:        $k^{(i)} \leftarrow \sum_{(\mathbf{x}, \mathbf{y}, \tilde{e}) \in \mathbf{z}_{U,E}^{(i)}} \mathbb{1}(\tilde{e} = 0)$
12:        $U^{(i)} \leftarrow U_{\text{Binom}}(k^{(i)}; |\mathbf{Z}_{U,E}^{(i)}|, \delta/\lceil \log_2 |\mathbf{Z}_{U,E}| \rceil)$
13:        **if** $U^{(i)} \leq \varepsilon$ **then**
14:           $\bar{i} \leftarrow \lceil (\underline{i}+\bar{i})/2 \rceil$
15:        **else**
16:           $\underline{i} \leftarrow \lceil (\underline{i}+\bar{i})/2 \rceil$
17:     $\tau_S \leftarrow \tau_S^{(i)}$
18:     $\hat{U} \leftarrow U^{(i)}$
19:     **return** $\tau_S, \hat{U}$

---

# E Unsupervised Selective Generation Algorithms (Certified)

---

**Algorithm 10** Unsupervised Selective Generator Learning with $\mathcal{R}_{\text{EM}}(\hat{S})$ Control [9]

---

1: **procedure** SG-EM($f_M$, $G$, $\mathbf{Z}_E$, $\mathbf{Z}_U$, $\varepsilon$, $\delta$)
2:     $\mathbf{Z}_{U,E} \leftarrow \mathbf{Z}_U \cup \mathbf{Z}_E$
3:     $\mathbf{Z}_{U,E} \leftarrow \text{SORT}_{f_M}(\mathbf{Z}_{U,E})$
4:     $(\underline{i}, \bar{i}) \leftarrow (1, |\mathbf{Z}_{U,E}|)$
5:     **for** $i = 1$ **to** $\lceil \log_2 |\mathbf{Z}_{U,E}| \rceil$ **do**
6:         $\tau_S^{(i)} \leftarrow f_M(\mathbf{x}_{\lceil (\underline{i}+\bar{i})/2 \rceil}, G(\mathbf{x}_{\lceil (\underline{i}+\bar{i})/2 \rceil}))$
7:        $\mathbf{Z}_{U,E}^{(i)} \leftarrow \{(\mathbf{x}, \mathbf{y}) \in \mathbf{Z}_{U,E} \mid f_M(\mathbf{x}, G(\mathbf{x})) \geq \tau_S^{(i)}\}$
8:        $k^{(i)} \leftarrow \sum_{(\mathbf{x}, \mathbf{y}) \in \mathbf{z}_{U,E}^{(i)}} \mathbb{1}(G(\mathbf{x}) \neq \mathbf{y})$
9:        $U^{(i)} \leftarrow U_{\text{Binom}}(k^{(i)}; |\mathbf{Z}_{U,E}^{(i)}|, \delta/\lceil \log_2 |\mathbf{Z}_{U,E}| \rceil)$
10:       **if** $U^{(i)} \leq \varepsilon$ **then**
11:          $\bar{i} \leftarrow \lceil (\underline{i}+\bar{i})/2 \rceil$
12:       **else**
13:          $\underline{i} \leftarrow \lceil (\underline{i}+\bar{i})/2 \rceil$
14:     $\tau_S \leftarrow \tau_S^{(i)}$
15:     $\hat{U} \leftarrow U^{(i)}$
16:     **return** $\tau_S, \hat{U}$

---

# F   Proof of Theorem 2

Let $C_\tau$ be a prediction set $C$ with a parameter $\tau$, $\mathcal{H}_\varepsilon := \{\tau \in \mathcal{H} \mid \mathcal{R}_{01}(C_\tau) > \varepsilon\}$, and $\tau^* := \inf \mathcal{H}_\varepsilon$, where $\mathcal{H}$ is finely-discretized non-negative real values. Then, we have

$$\mathbb{P}\Big\{\mathcal{R}_{01}(\mathcal{A}_{\mathrm{Binom}}(\mathbf{Z})) > \varepsilon\Big\} \leq \mathbb{P}\Big\{\exists \tau \in \mathcal{H}_\varepsilon, U_{\mathrm{Binom}}(k_\tau; n, \delta) \leq \varepsilon\Big\}$$

$$\leq \mathbb{P}\Big\{U_{\mathrm{Binom}}(k_{\tau^*}; n, \delta) \leq \varepsilon\Big\} \tag{11}$$

$$\leq \mathbb{P}\Big\{\mathcal{R}_{01}(C_{\tau^*}) > \varepsilon \wedge U_{\mathrm{Binom}}(k_{\tau^*}; n, \delta) \leq \varepsilon\Big\}$$

$$\leq \mathbb{P}\Big\{\mathcal{R}_{01}(C_{\tau^*}) > U_{\mathrm{Binom}}(k_{\tau^*}; n, \delta)\Big\} \leq \delta, \tag{12}$$

where the last equality in (11) holds as $\mathbb{1}(\mathbf{y} \notin C_\tau(\mathbf{x}))$ and $U_{\mathrm{B}}$ are non-decreasing in $\tau$ (*i.e.,* Lemma 2 in [34]) and the last inequality in (12) is due to the property of the binomial tail bound $U_{\mathrm{Binom}}$.

# G   Proof of Lemma 2

Since (E) in (2) is decomposed into three terms in Lemma 1, we first find upper bounds on each of the terms and take the union bound as follows. This will return a single upper bound on (E) in (2), which we denote $U_{\mathrm{SSL}}$.

**FER Bound.** First, recall that

$$\mathcal{R}_{\mathrm{FER}}(\hat{E}) := \mathbb{P}_{\mathcal{D}_{\hat{S}}}\{e = 0 \wedge G(\mathbf{x}) \in \hat{E}(\mathbf{y})\}.$$

Learning $\hat{E}$ via $\mathcal{A}_{\mathrm{FER}}$ is equivalent to the PAC prediction set learning algorithm that considers the optimization problem in (10), where the indicator loss is $\ell_{01}(\hat{E}, \mathbf{x}, \mathbf{y}, e) := \mathbb{1}(e = 0 \wedge G(\mathbf{x}) \in \hat{E}(\mathbf{y}))$ and the target model is the entailment scoring function $f_E$. Therefore, by Theorem 2, for any $n_E := |\mathbf{Z}_E|$, we have

$$\mathbb{P}_{\mathbf{Z}_E}\Big\{\mathcal{R}_{\mathrm{FER}}(\hat{E}) \leq \varepsilon_E\Big\} = \sum_{m=1}^{n_E} \mathbb{P}_{\mathbf{Z}_E}\Big\{\mathcal{R}_{\mathrm{FER}}(\hat{E}) \leq \varepsilon_E \mid |\hat{\mathbf{Z}}_E| = m\Big\} \cdot \mathbb{P}_{\mathbf{Z}_E}\Big\{|\hat{\mathbf{Z}}_E| = m\Big\}$$

$$\geq \sum_{m=1}^{n_E} (1 - \delta'_E/2) \cdot \mathbb{P}_{\mathbf{Z}_E}\Big\{|\hat{\mathbf{Z}}_E| = m\Big\} \tag{13}$$

$$= 1 - \delta'_E/2. \tag{14}$$

Note that (13) holds as the PAC guarantee for conformal prediction holds for any number of samples.

The same bound holds with respect to $\mathbf{Z}$. Specifically, letting $\ell_{\mathrm{FER}}(\mathbf{Z}_E, \mathbf{Z}_U) := \mathbb{1}(\mathcal{R}_{\mathrm{FER}}(\hat{E}) \leq \varepsilon_E)$, we have

$$\mathbb{P}_{\mathbf{Z}}\Big\{\mathcal{R}_{\mathrm{FER}}(\hat{E}) \leq \varepsilon_E\Big\} = \int \ell_{\mathrm{FER}}(\mathbf{Z}_E, \mathbf{Z}_U) \, \mathrm{d}\mathbb{P}(\mathbf{Z})$$

$$= \int \ell_{\mathrm{FER}}(\mathbf{Z}_E, \mathbf{Z}_U) \, \mathrm{d}\mathbb{P}(\mathbf{Z}_E) \mathrm{d}\mathbb{P}(\mathbf{Z}_U)$$

$$\geq \int (1 - \delta'_E/2) d\mathbb{P}(\mathbf{Z}_U)$$

$$= 1 - \delta'_E/2, \tag{15}$$

where the second equality holds due to the i.i.d. assumption on the calibration data and the inequality holds due to (14).

**FNER Bound.** Recall

$$\mathcal{R}_{\mathrm{FNER}}(\hat{E}) := \mathbb{P}_{\mathcal{D}_{\hat{S}}}\{e = 1 \wedge \hat{e} = 0\}.$$

Since our goal is to upper-bound $-\mathcal{R}_{\text{FNER}}(\hat{E})$, we consider a lower bound $\mathcal{R}_{\text{FNER}}(\hat{E})$ as follows for any $n_E := |\mathbf{Z}_E|$:

$$\mathbb{P}_{\mathbf{Z}_E}\left\{\mathcal{R}_{\text{FNER}}(\hat{E}) \geq L_{\text{binom}}(\hat{k}; |\hat{\mathbf{Z}}_E|, \delta'_E/2)\right\}$$

$$= \sum_{m=1}^{n_E} \mathbb{P}_{\mathbf{Z}_E}\left\{\mathcal{R}_{\text{FNER}}(\hat{E}) \geq L_{\text{binom}}(\hat{k}; |\hat{\mathbf{Z}}_E|, \delta'_E/2) \,\Big|\, |\hat{\mathbf{Z}}_E| = m\right\} \cdot \mathbb{P}_{\mathbf{Z}_E}\{|\hat{\mathbf{Z}}_E| = m\}$$

$$\geq \sum_{m=1}^{n_E} (1 - \delta'_E/2) \cdot \mathbb{P}_{\mathbf{Z}_E}\{|\hat{\mathbf{Z}}_E| = m\},$$

$$= 1 - \delta'_E/2 \tag{16}$$

where the inequality holds due to the binomial tail bound. The same bound holds when the probability is taken over $\mathbf{Z}$. First, let

$$\ell_{\text{FNER}}(\mathbf{Z}_E, \mathbf{Z}_U) := \mathbb{1}\left(\mathcal{R}_{\text{FNER}}(\hat{E}) \geq L_{\text{Binom}}(\hat{k}; |\hat{\mathbf{Z}}_E|, \delta'_E/2)\right).$$

Then,

$$\mathbb{P}_{\mathbf{Z}}\{\mathcal{R}_{\text{FNER}}(\hat{E}) \geq L_{\text{Binom}}(\hat{k}; |\hat{\mathbf{Z}}_E|, \delta'_E/2)\} = \int \ell_{\text{FNER}}(\mathbf{Z}_E, \mathbf{Z}_U) d\mathbb{P}(\mathbf{Z})$$

$$= \int \ell_{\text{FNER}}(\mathbf{Z}_E, \mathbf{Z}_U) d\mathbb{P}(\mathbf{Z}_E) d\mathbb{P}(\mathbf{Z}_U)$$

$$\geq \int (1 - \delta'_E/2) d\mathbb{P}(\mathbf{Z}_U)$$

$$= 1 - \delta'_E/2, \tag{17}$$

where the second equality holds due to the i.i.d. assumption and the inequality holds due to (16).

**NER Bound.** Recall

$$\mathcal{R}_{\text{NER}}(\hat{E}) := \mathbb{P}_{\mathcal{D}_{\hat{S}}}\{\hat{e} = 0\} = \mathbb{P}_{\mathcal{D}_{\hat{S}}}\{G(\mathbf{x}) \notin \hat{E}(\mathbf{y})\}.$$

Then, we upper bound $\mathcal{R}_{\text{NER}}(\hat{E})$ as follows for any $n_U := |\mathbf{Z}_U|$:

$$\mathbb{P}_{\mathbf{Z}_U}\left\{\mathcal{R}_{\text{NER}}(\hat{E}) \leq U_{\text{Binom}}(\hat{l}; |\hat{\mathbf{Z}}_U|, \delta'_S)\right\}$$

$$= \sum_{m=1}^{n_U} \mathbb{P}_{\mathbf{Z}_U}\left\{\mathcal{R}_{\text{NER}}(\hat{E}) \leq U_{\text{Binom}}(\hat{l}; |\hat{\mathbf{Z}}_U|, \delta'_S) \,\Big|\, |\hat{\mathbf{Z}}_U| = m\right\} \cdot \mathbb{P}_{\mathbf{Z}_U}\{|\hat{\mathbf{Z}}_U| = m\}$$

$$\geq \sum_{m=1}^{n_U} (1 - \delta'_S) \cdot \mathbb{P}_{\mathbf{Z}_U}\{|\hat{\mathbf{Z}}_U| = m\}$$

$$= 1 - \delta'_S, \tag{18}$$

where the inequality holds due to the binomial tail bound. Again, the same bound holds when the probability is taken over $\mathbf{Z}$. First, let

$$\ell_{\text{NER}}(\mathbf{Z}_E, \mathbf{Z}_U) := \mathbb{1}\left(\mathcal{R}_{\text{NER}}(\hat{E}) \leq U_{\text{Binom}}(\hat{l}; |\hat{\mathbf{Z}}_U|, \delta'_S)\right)$$

Then,

$$\mathbb{P}_{\mathbf{Z}}\{\mathcal{R}_{\text{NER}}(\hat{E}) \leq U_{\text{Binom}}(\hat{l}; |\hat{\mathbf{Z}}_U|, \delta'_S)\} = \int \ell_{\text{NER}}(\mathbf{Z}_E, \mathbf{Z}_U) d\mathbb{P}(\mathbf{Z})$$

$$= \int \ell_{\text{NER}}(\mathbf{Z}_E, \mathbf{Z}_U) d\mathbb{P}(\mathbf{Z}_U) d\mathbb{P}(\mathbf{Z}_E)$$

$$\geq \int (1 - \delta'_S) d\mathbb{P}(\mathbf{Z}_E)$$

$$= 1 - \delta'_S, \tag{19}$$

where the inequality holds due to (18).

Finally, taking the union bound of (15), (17), and (19) completes the proof.

# H  Proof of Lemma 3

Let $U_{\text{SSL}}^{(i)}$ be $U_{\text{SSL}}$ for the $i$-th candidate of $\varepsilon_E$ in Algorithm 3. Due to Lemma 2, the following holds:

$$\mathbb{P}_{\mathbf{Z}}\big\{\mathbb{P}_{\mathcal{D}_{\hat{S}}}\{e=0\} > U_{\text{SSL}}^{(i)})\big\} \leq (\delta'_E + \delta'_S)/Q.$$

Since $U_{\text{SSL}}^{\text{OPT}} = \min_{i \in [Q]} U_{\text{SSL}}^{(i)}$, we have

$$\mathbb{P}_{\mathbf{Z}}\big\{\mathbb{P}_{\mathcal{D}_{\hat{S}}}\{e=0\} > U_{\text{SSL}}^{\text{OPT}}\big\} \leq \mathbb{P}_{\mathbf{Z}}\big\{\exists\, i \in \{1, \ldots, Q\}, \mathbb{P}_{\mathcal{D}_{\hat{S}}}\{e=0\} > U_{\text{SSL}}^{(i)}\big\}$$

$$\leq \sum_{i=1}^{Q} \mathbb{P}_{\mathbf{Z}}\big\{\mathbb{P}_{\mathcal{D}_{\hat{S}}}\{e=0\} > U_{\text{SSL}}^{(i)}\big\}$$

$$\leq \delta'_E + \delta'_S,$$

where the second inequality is due to a union bound. This completes the proof.

# I  Proof of Theorem 1

Let $\mathcal{H}$ be the calibration set-dependent hypothesis space of selective generators, where $n_{\mathcal{H}} := |\mathcal{H}|$ is always calibration set independent. Letting $U^{(i)}$ be the FDR-E bound computed given the $i$-th selective generator $S_i$ in $\mathcal{H}$, we first describe how to derive an upper bound of the FDR-E for a given hypothesis $S_i$.

Since an upper bound of (E) in (2) is proved in Lemma 3, the remaining parts are (i) to derive upper bounds on the others and (ii) to take the union bound. For proportions of the visibility of textual entailment labels, *i.e.,* (B) and (D) in (2), and the FDR-E for the supervised case only using entailment-labeled examples, *i.e.,* (C) in (2), the followings hold due to the binomial tail bound:

$$\mathbb{P}_{\mathbf{Z}}\Big\{\mathbb{P}_{\mathcal{D}_{S_i}}\{v=1\} \leq \underbrace{U_{\text{Binom}}\big(|\hat{\mathbf{Z}}_E|; |\hat{\mathbf{Z}}_E| + |\hat{\mathbf{Z}}_U|, \delta_W/(2 \times |\mathcal{H}|)\big)}_{:=w_{\text{SL}}^{(i)}}\Big\} \geq 1 - \delta_W/(2 \times |\mathcal{H}|);$$

$$\mathbb{P}_{\mathbf{Z}}\Big\{\mathbb{P}_{\mathcal{D}_{S_i}}\{v=0\} \leq \underbrace{U_{\text{Binom}}\big(|\hat{\mathbf{Z}}_U|; |\hat{\mathbf{Z}}_E| + |\hat{\mathbf{Z}}_U|, \delta_W/(2 \times |\mathcal{H}|)\big)}_{:=w_{\text{SSL}}^{(i)}}\Big\} \geq 1 - \delta_W/(2 \times |\mathcal{H}|);$$

$$\mathbb{P}_{\mathbf{Z}}\Big\{\mathbb{P}_{\mathcal{D}_{S_i}}\{e=0\} \leq \underbrace{U_{\text{Binom}}\big(|\hat{\mathbf{Z}}_E^{e=0}|; |\hat{\mathbf{Z}}_E|, \delta_S/(2 \times |\mathcal{H}|)\big)}_{:=U_{\text{SL}}^{(i)}}\Big\} \geq 1 - \delta_S/(2 \times |\mathcal{H}|),$$

where $\hat{\mathbf{Z}}_E$ and $\hat{\mathbf{Z}}_U$ are defined same as Lemma 2 does, and $\hat{\mathbf{Z}}_E^{e=0} := \{(\mathbf{x}, \mathbf{y}, e) \in \hat{\mathbf{Z}}_E \mid e = 0\}$. Note that the binomial tail bound is applied to filtered sets by the given selective generator (*e.g.,* $\hat{\mathbf{Z}}_E$), but we can use the same bound for the non-filtered set $\mathbf{Z}$, by using the same marginalization technique over the size of a filtered set, as in, *e.g.,* (15).

Thus, by taking the union bound along with Lemma 3 when $\delta'_E = \delta_E$ and $\delta'_S = \delta_S/2$,

$$\mathbb{P}_{\mathbf{Z}}\big\{\mathcal{R}_E(S_i) \leq U^{(i)}\big\} \geq 1 - (\delta_E + \delta_S + \delta_W)/|\mathcal{H}|, \tag{20}$$

where $U_i := w_{\text{SL}}^{(i)} U_{\text{SL}}^{(i)} + w_{\text{SSL}}^{(i)} U_{\text{SSL}}^{\text{OPT}^{(i)}}$ is the computed FDR-E bound a given selective generator $S_i$. Here, $U_{\text{SSL}}^{\text{OPT}^{(i)}}$ refers to the smallest FDR-E bound of (E) in (2) given the $i$-th selective generator.

Since (20) holds for all $S_i \in \mathcal{H}$, and the final bound $\hat{U}$ is chosen among them, this completes the proof by taking an union bound, *i.e.,*

$$\mathbb{P}_{\mathbf{Z}}\left\{\mathcal{R}_E(\hat{S}) > \hat{U}\right\} \le \mathbb{P}_{\mathbf{Z}}\left\{\exists S_i \in \mathcal{H}, \mathcal{R}_E(S_i) > U_i\right\}$$

$$= \sum_{k=1}^{n_{\mathcal{H}}} d\mathbb{P}_{\mathbf{Z}}\left\{\exists S_i \in \mathcal{H}, \mathcal{R}_E(S_i) > U_i, |\mathcal{H}| = k\right\}$$

$$= \sum_{k=1}^{n_{\mathcal{H}}} \mathbb{P}_{\mathbf{Z}}\left\{\exists S_i \in \mathcal{H}, \mathcal{R}_E(S_i) > U_i \mid |\mathcal{H}| = k\right\}\mathbb{P}_{\mathbf{Z}}\left\{|\mathcal{H}| = k\right\}$$

$$\le \sum_{k=1}^{n_{\mathcal{H}}}\sum_{i=1}^{k} \mathbb{P}_{\mathbf{Z}}\left\{\mathcal{R}_E(S_i) > U_i \mid |\mathcal{H}| = k\right\}\mathbb{P}_{\mathbf{Z}}\left\{|\mathcal{H}| = k\right\}$$

$$\le \sum_{k=1}^{n_{\mathcal{H}}}\sum_{i=1}^{k} \left(\frac{\delta_E + \delta_S + \delta_W}{k}\right)\mathbb{P}_{\mathbf{Z}}\left\{|\mathcal{H}| = k\right\}$$

$$= \delta_E + \delta_S + \delta_W.$$

## J  Proof of Lemma 4

We say $f_M$ is perfectly calibrated with respect to $\mathcal{D}$, $G$, $E_{\text{true}}$ if

$$\mathbb{P}_{\mathcal{D}}\{G(\mathbf{x}) \in E_{\text{true}}(\mathbf{y}) \mid f_M(\mathbf{x}, G(\mathbf{x})) = t\}) = t, \forall t. \tag{21}$$

The true discovery rate with respect to $E_{\text{true}}$ conditioned on $f_M(\mathbf{x}, G(\mathbf{x})) \ge \tau_S$, *i.e.*, $1 - \text{FDR-E}$, is as follows:

$$\mathbb{P}\{G(\mathbf{x}) \in E_{\text{true}}(\mathbf{y}) \mid f_M(\mathbf{x}, G(\mathbf{x})) \ge \tau_S\}$$

$$= \frac{\int_{\tau_S}^{1} \mathbb{P}\{G(\mathbf{x}) \in E_{\text{true}}(\mathbf{y}) \mid f_M(\mathbf{x}, G(\mathbf{x})) = t\}\mathbb{P}\{f_M(\mathbf{x}, G(\mathbf{x})) = t\}dt}{\int_{\tau_S}^{1} \mathbb{P}\{f_M(\mathbf{x}, G(\mathbf{x})) = t\}dt}$$

$$= \frac{\int_{\tau_S}^{1} t\mathbb{P}\{f_M(\mathbf{x}, G(\mathbf{x})) = t\}dt}{\int_{\tau_S}^{1} \mathbb{P}\{f_M(\mathbf{x}, G(\mathbf{x})) = t\}dt}, \tag{22}$$

where and (22) holds as $f_M$ is perfectly calibrated, *i.e.*, (21).

Letting $h(t) := \mathbb{P}\{f_M(\mathbf{x}, G(\mathbf{x})) = t\}$, $H(t) := \int_t^1 h(t')dt'$, $i(t) := t\mathbb{P}\{f_M(\mathbf{x}, G(\mathbf{x})) = t\}$, and $I(t) := \int_t^1 i(t')dt'$, since we have $\tau_S \le \frac{\int_{\tau_S}^1 t\mathbb{P}\{f_M(\mathbf{x}, G(\mathbf{x}))=t\}dt}{\int_{\tau_S}^1 \mathbb{P}\{f_M(\mathbf{x}, G(\mathbf{x}))=t\}dt} \le 1$, the following holds:

$$I(1) - I(\tau_S) \ge \tau_S(H(1) - H(\tau_S)).$$

Therefore,

$$\frac{d}{d\tau_S}\mathbb{P}\{G(\mathbf{x}) \in E_{\text{true}}(\mathbf{y}) \mid f_M(\mathbf{x}, G(\mathbf{x})) \ge \tau_S\} = \frac{d}{d\tau_S}\left\{\frac{I(1) - I(\tau_S)}{H(1) - H(\tau_S)}\right\}$$

$$= \frac{-h(\tau_S)\Big[\tau_S(H(1) - H(\tau_S)) - (I(1) - I(\tau_S))\Big]}{(H(1) - H(\tau_S))^2}$$

$$\ge 0.$$

This completes the proof.

Note that the classification problem can be reduced from the special case, *i.e.*, $E_{\text{true}}(y) := E_{\text{EM}}(y)$, where $\mathcal{Y} := \mathcal{W}$ and $E_{\text{EM}}(y) := \{y\} = \arg\max_{w \in \mathcal{W}} \mathbb{P}(Y = w \mid \mathbf{X} = \mathbf{x})$.

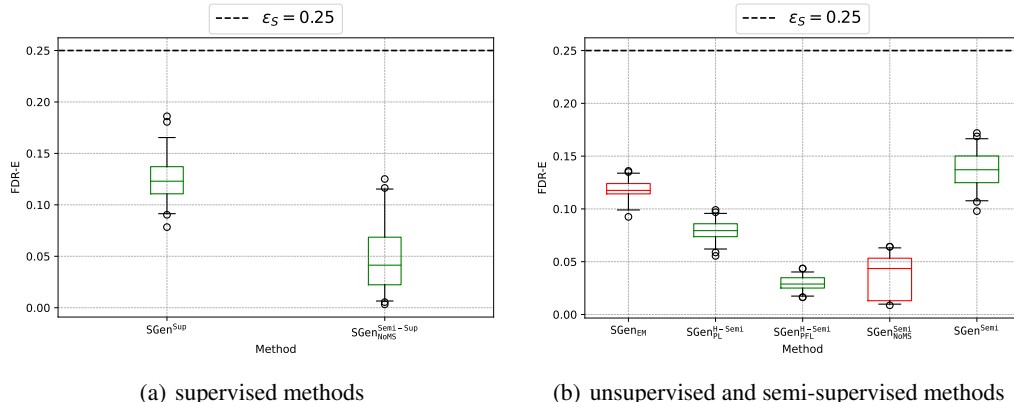

(a) supervised methods      (b) unsupervised and semi-supervised methods

Figure 4: FDR-E box plots of methods for GPT-3.5-turbo. We randomly split the calibration ad test set 100 times for box plots. For supervised methods (a), we use all entailment labels, *i.e.*, $|\mathbf{Z}_E| = |\mathbf{Z}_E^{\text{cal}}|$. For (b), which includes an unsupervised method ($\texttt{SGen}_{\texttt{EM}}$) and semi-supervised methods, we use $|\mathbf{Z}_E| = 0.75|\mathbf{Z}_E^{\text{cal}}|$. All methods except for $\texttt{SGen}^{\texttt{Semi}}$ use $f_{M_1}$ as a score function. The methods that do not control $\varepsilon_S$ FDR-E in learning at least once are drawn using red boxes but otherwise using green boxes in Figure 4(a) and Figure 4(b). We draw the whisker plot to indicate $100\delta\%$ and $100(1-\delta)\%$ quantiles. In both (a) and (b) with green boxes, as the top of the whisker is below of the dotted line, we can see that the FDR-E is well controlled with probability at least $\delta$, *i.e.,* they satisfy the PAC guarantee. The numbers of iterations that satisfy $\varepsilon_S$ FDR-E in learning while running 100 iterations are (a) $\texttt{SGen}_{\texttt{EM}}= 0$, $\texttt{SGen}^{\texttt{Sup}}= 100$, $\texttt{SGen}_{\texttt{NoMS}}^{\texttt{Semi-Sup}}= 100$ and (b) $\texttt{SGen}_{\texttt{PL}}^{\texttt{H-Semi}}= 100$, $\texttt{SGen}_{\texttt{PFL}}^{\texttt{H-Semi}}= 100$, $\texttt{SGen}_{\texttt{NoMS}}^{\texttt{Semi}}= 18$, $\texttt{SGen}^{\texttt{Semi}}= 100$.

## K    Additional Experiments

Table 3: Comparison results of fully supervised methods. Here, we use all entailment labels, *i.e.,* $|\mathbf{Z}_E| = |\mathbf{Z}_E^{\text{cal}}|$ for GPT-3.5-turbo and Alpaca-7B. The best results are highlighted in bold, results from methods that do not satisfy desired FDR-E guarantee are underlined. In GPT-3.5-turbo and Alpaca-7B, the best efficiency values among methods that satisfy a desired FDR-E guarantee are $0.7535$ and $0.2959$, respectively, which serve as the best achievable efficiency results of semi-supervised methods.

| | Models | GPT-3.5-turbo | | Alpaca-7B | |
|---|---|---|---|---|---|
| | Methods | $\texttt{SGen}^{\texttt{Sup}}$ | $\texttt{SGen}_{\texttt{NoMS}}^{\texttt{Semi-Sup}}$ | $\texttt{SGen}^{\texttt{Sup}}$ | $\texttt{SGen}_{\texttt{NoMS}}^{\texttt{Semi-Sup}}$ |
| $f_{M_1}$ | FDR-E | 0.1697 | 0.1066 | 0.0400 | 0.0231 |
| | efficiency | 0.6474 | 0.4657 | 0.1769 | 0.0922 |
| $f_{M_2}$ | FDR-E | 0.2209 | 0.0914 | 0.0983 | 0.0827 |
| | efficiency | 0.8596 | 0.5408 | 0.4149 | 0.3675 |
| average efficiency | | 0.7535 | 0.5033 | 0.2959 | – |

Table 4: Comparison results of semi-supervised methods. Here, $|\mathbf{Z}_U| = 10K$ for GPT-3.5-turbo and Alpaca-7B. The best results are highlighted in **bold** and results from methods that do not satisfy desired FDR-E guarantee are underlined. We used QA2D dataset, filtered with only SQuAD, where human transformed QA sentences exist. $\varepsilon = 0.15$.

| Models | | GPT-3.5-turbo | | | | |
|---|---|---|---|---|---|---|
| Methods | | Heuristic | | Certified | | |
| | | $\text{SGen}_{\text{PL}}^{\text{H-Semi}}$ | $\text{SGen}_{\text{PFL}}^{\text{H-Semi}}$ | $\text{SGen}_{\text{EM}}$ | $\text{SGen}_{\text{NoMS}}^{\text{Semi}}$ | $\text{SGen}^{\text{Semi}}$ |
| $f_{M_1}$ | FDR-E | 0.0000 | 0.0000 | 0.0213 | 0.0962 | 0.0918 |
| | efficiency | 0.0387 | 0.0227 | 0.4775 | 0.8608 | 0.8502 |
| $f_{M_2}$ | FDR-E | 0.0053 | 0.0039 | 0.0831 | 0.0169 | 0.0918 |
| | efficiency | 0.1300 | 0.1025 | 0.4862 | 0.2156 | 0.8502 |
| average efficiency | | 0.0844 | 0.0626 | – | 0.5382 | 0.8502 |

Table 5: Comparison results of fully supervised methods. Here, we use all entailment labels, *i.e.,* $|\mathbf{Z}_E| = |\mathbf{Z}_E^{\text{cal}}|$ for GPT-3.5-turbo and Alpaca-7B. The best results are highlighted in bold, results from methods that do not satisfy desired FDR-E guarantee are underlined. We used QA2D dataset, filtered with only SQuAD, where human transformed QA sentences exist. $\varepsilon = 0.15$.

| Models | | GPT-3.5-turbo | |
|---|---|---|---|
| Methods | | $\text{SGen}^{\text{Sup}}$ | $\text{SGen}_{\text{NoMS}}^{\text{Semi-Sup}}$ |
| $f_{M_1}$ | FDR-E | 0.1116 | 0.0454 |
| | efficiency | 0.8956 | 0.6525 |
| $f_{M_2}$ | FDR-E | 0.0459 | 0.0082 |
| | efficiency | 0.3185 | 0.1532 |
| average efficiency | | 0.6071 | 0.4029 |

