# OpenReview forum: "Selective Generation for Controllable Language Models"
_NeurIPS.cc/2024/Conference — NeurIPS 2024 spotlight_

### Official Review · Reviewer_UCLF · 2024-07-11

**Soundness:** 3
**Presentation:** 3
**Contribution:** 3
**Rating:** 6
**Confidence:** 3

**Summary:**

This paper introduces Neuro-Selective Entailing-Generation (NSEGen), a novel approach to enhance the trustworthiness of generative language models. The method extends selective classification to language generation tasks, utilizing textual entailment to measure semantic correctness between generated and true answers. NSEGen employs a semi-supervised method that leverages both labeled and unlabeled data by learning an entailment set, addressing the challenge of expensive entailment labeling. The authors introduce neuro-selection functions to optimize feature selection for minimizing false discovery rate and provide theoretical guarantees on false discovery rate control. The approach aims to control the entailment-based false discovery rate while maximizing selection efficiency.

**Strengths:**

- The paper introduces a new framework for improving GLM trustworthiness that addresses limitations of existing methods. This is particularly important because it tackles the metric misalignment issue in GLMs, where conventional metrics like exact match fail to capture semantic correctness. By introducing selective generation and leveraging textual entailment, the authors provide a more nuanced and accurate way to evaluate language model outputs.
- The approach is supported by thorough theoretical analysis and guarantees. The authors provide a detailed proof for the correctness guarantee of their algorithm (Theorem 1) and establish conditions for achieving monotonicity in precision (Lemma 3).

**Weaknesses:**

- The method introduces several new components and parameters, which may make it challenging to implement and tune. For instance, the algorithm involves learning an entailment set, designing neuro-selection functions, and tuning multiple parameters. This complexity could make it difficult for practitioners to adopt the method.
- Experiments are conducted only on Natural Question with two models (GPT-3.5 and Alpaca-7B). While these experiments do demonstrate the method's effectiveness, they leave open questions about its generalizability. Especially if the method claims to adapt uncertainty learning methods to generation tasks. It would be valuable to see how the method performs on a wider range of language generation tasks (e.g., summarization, translation, or open-ended text generation and so on).

**Questions:**

N/A

---

> ### Author Rebuttal · Authors · 2024-08-06
>
> > The method introduces several new components and parameters, which may make it challenging to implement and tune. For instance, the algorithm involves learning an entailment set, designing neuro-selection functions, and tuning multiple parameters. This complexity could make it difficult for practitioners to adopt the method.
> - We thank you for valuable comments on practical use of our algorithm. Even though the algorithm may seem complex at first glance, we want to highlight that there are only a few user-specified parameters that the user should take care of to run our algorithm, all of which have intuitive interpretations.
> - Here, we provide (1) a detailed description of how to choose the user-specified parameters and (2) additional guidelines for the practical use of our method. First, we would like to start with (2). The following is a list of configurations that practitioners need to choose before running our algorithm, which may vary based on the language generation problem they aim to solve. Since we assume the configurations are easy for practitioners to choose, we distinguish them from what we call “user-specified parameters”.
>    * Generator $G$: Target GLM that we aim to control the rate of hallucinations for the specific language generation task
>    * $Z_E, Z_U$: Calibration data with and without labels on textual entailment
>    * $f_{M_1}, f_{M_2}$: Language scoring functions to quantify the uncertainty of the generated sequence on a given problem
>    * $f_E$: Entailment classifier to predict the textual entailment
> - The remainder provides a detailed description of choosing the user-specified parameters: $\epsilon_S, \epsilon_E$, and $\delta(=\delta_E+\delta_S)$.
>    * For $\epsilon_E$, we newly propose an automatic $\epsilon_E$-searching algorithm for convenience. We hope to add this algorithm in our final manuscript, since we believe this would make our algorithm more accessible to users.
>       * $\epsilon_E$: The target level of mislabeling error made in the pseudo-labeling process. Unlike $\epsilon_S$ which specifies the target rate of hallucinations (FDR-E), we thought that letting the user choose $\epsilon_E$ may not be easy. Hence, we propose a $\epsilon_E$-searching algorithm which automatically chooses $\epsilon_E$ among possible candidates. Specifically, we find the smallest $\epsilon_E$ that returns a non-vacuous entailment set among candidates. If a detailed algorithmic description or a modified version of proof is needed, we will additionally provide them via an anonymized file.
>    * The following are guidelines for choosing the remaining user-specified parameters. Although they may not seem straightforward to choose, they have intuitive interpretations that make them easy to specify.
>       * $\epsilon_S$: As mentioned above, $\epsilon_S$ is the target rate of hallucinations (FDR-E) chosen by the user, which is straightforward. Based on the type of language generation problem or the user’s preference between the rate of hallucinations and the number of abstained answers, he/she can select $\epsilon_S$.
>       * $\delta ( = \delta_S + \delta_E )$: $\delta$ is related to the confidence of the guarantee on FDR-E. Specifically, irrespective of which calibration set $(Z_E, Z_U)$ we use, our algorithm controls the FDR-E with probability $1 - \delta$ (Theorem 1). This can be chosen based on the degree of confidence that the user thinks appropriate (e.g., 90% of confidence).
>
> > Experiments are conducted only on Natural Question with two models (GPT-3.5 and Alpaca-7B). While these experiments do demonstrate the method's effectiveness, they leave open questions about its generalizability. Especially if the method claims to adapt uncertainty learning methods to generation tasks. It would be valuable to see how the method performs on a wider range of language generation tasks (e.g., summarization, translation, or open-ended text generation and so on).
> - Thanks for raising concerns on the generalizability of our method. We think it would be great to verify and further generalize the applicability of our method to a wider range of language generation tasks and these would be very interesting future work. Although we have conducted an experiment only on the open-ended QA problem, we want to note that our problem and algorithm are designed in a general manner.
> - As of generalizability, one of technical issues is concerned with the choice of the entailment classifier. Specifically, the success of our semi-supervised version, a label-efficient version which exploits the unlabeled data set via pseudo-labeling, depends on whether we have access to an accurate entailment set for the given language generation problem. Specifically, as can be seen in Algorithm 1, the accuracy of the entailment set depends on the performance of the textual entailment classifier. Therefore, the applicability of our algorithm depends on the choice of the entailment classifier and its quality on the specific type of language-generation problem we consider. In this paper, we consider the transformer-based textual entailment classifier which is originally trained on the natural language inference (NLI) dataset. Since the NLI dataset consists of pairs of premises and hypotheses in a declarative form of moderate length, the entailment classifier would not perform well on pairs of long sequences. For such cases where the pretrained entailment classifier provides inaccurate predictions, extra calibration data are needed to train an entailment classifier from scratch [R1], or to finetune the base model [R2].
>    * [R1] Yu Gui, Ying Jin, and Zhimei Ren. “Conformal Alignment: Knowing When to Trust Foundation Models with Guarantees.” ArXiv, 2024.
>
>    * [R2] Christopher Mohri, et al. “Learning to Reject with a Fixed Predictor: Application to Decontextualization.” ICLR, 2024.

---

### Official Review · Reviewer_KeFc · 2024-07-12

**Soundness:** 3
**Presentation:** 4
**Contribution:** 4
**Rating:** 8
**Confidence:** 4

**Summary:**

The paper investigates the trustworthiness of generative language models in critical decision-making systems, identifying deficiencies in current uncertainty learning methods, such as selective classification and conformal prediction, which fail to address metric misalignment in GLMs. The authors propose an entailment-based false discovery rate  by leveraging logical entailment to define the relationship between true and generated answers. They introduce a supervised selective generation approach using entailment labels to control the FDR and a semi-supervised method to mitigate the high cost of obtaining these labels. This method employs an entailment set for pseudo-labeling and neuro-selection functions to enhance data space learning, reducing FDR. The neuro-selective entailing-generation algorithm  is theoretically validated, meeting the PAC guarantee on the desired FDR. NSeGen demonstrates improved efficiency and reliability in achieving a desired FDR level compared to baseline models, highlighting its practical applications.

**Strengths:**

The work works to provide good grounding of both the problem and proposed path to the solution. With the proposed path, the authors work well to highlight the deficiencies both with current approaches, as well as their chosen approach and then incorporate fixes that make the deficiencies less of a problem. The work to do psuedo labelling and also deal with entailment is important as well as showing the theoretical guarantees.

**Weaknesses:**

1. There is a bit of repetitiveness in the Experiment section when describing the GLMs and Datasets.
2. It should be made clear that your approach deals with the generation results from the underlying model. For example, when reading table 1 earlier I thought you were comparing Alpaca to your method, but later got it as you have different ways to deal with the selective generation.
3. It would be great to get a practical understanding of how much data would be needed to train a good enough selective algorithm. This is important especially in low resource scenarios (both language and application areas) as the space of application of where such an algorithm such as yours might be used is highly likely in places were there is not much data. So ablation studies on sizes of the training set would have been great.

**Questions:**

See my weaknesses. esp.

3. It would be great to get a practical understanding of how much data would be needed to train a good enough selective algorithm. This is important especially in low resource scenarios (both language and application areas) as the space of application of where such an algorithm such as yours might be used is highly likely in places were there is not much data. So ablation

**Limitations:**

The work covers their limitations well and this was very appreciated. A practical guide for researchers might have been great in terms of knowing how much data is needed to help the psuedo labelling.

---

> ### Author Rebuttal · Authors · 2024-08-06
>
> > There is a bit of repetitiveness in the Experiment section when describing the GLMs and Datasets.
> * Thank you for your suggestion. As you mentioned, there are redundancies in our descriptions of GLMs and datasets. The following is our revised version in a concise but detailed manner, and we will update it in our final manuscript.
>     * Models and Datasets. We use two large language models (LLMs), GPT-3.5-Turbo and Alpaca-7B, for language generation. Specifically, we consider the open-ended QA task, where the model is prompted to generate the answer in a declarative form given a question. To validate our method and its theoretical guarantee on controlling FDR-E, we create a dataset on textual entailment using the Natural Questions (NQ) dataset [43] for each GLM. Based on the transformation method by [44] that converts the question and answer pair in QA dataset into a declarative form, we manually labeled textual entailment by letting the generated sequence as the premise and the reference answer in declarative form as hypothesis. Approximately 7.3k (7,374) and 3k (3,000) samples are labeled for GPT-3.5-Turbo and Alpaca-7B, respectively, and both are split into calibration data at an 8:2 ratio. For semi-supervised learning algorithms that exploit unlabeled data, at most 27k and 10k unlabeled samples are used to train a selective generator, varying its size. Besides, semi-supervised learning algorithms use only 75% of the labeled calibration data compared to what is used by supervised methods.
>
> > It should be made clear that your approach deals with the generation results from the underlying model. For example, when reading table 1 earlier I thought you were comparing Alpaca to your method, but later got it as you have different ways to deal with the selective generation.
> * Thank you for your suggestion. We agree that there should be a clarification that our approach depends on generation results of the underlying GLM we consider, even with the same dataset. We will update it in our final manuscript, which is the revised version of descriptions on GLMs and datasets from the above question.
>
> > It would be great to get a practical understanding of how much data would be needed to train a good enough selective algorithm. This is important especially in low resource scenarios (both language and application areas) as the space of application of where such an algorithm such as yours might be used is highly likely in places where there is not much data. So ablation studies on sizes of the training set would have been great.
> * Thanks for your advice. We conducted the ablation study on NQ dataset while maintaining the ratio of $|\mathbf{Z}_E|$ and $|\mathbf{Z}_U|$. We simply multiplied the sizes of $|\mathbf{Z}_E|$ and $|\mathbf{Z}_U|$ by 1, 0.7, 0.5, 0.3, and 0.1. To set initial data sizes of each model similar in ratio, the sizes of GPT-3.5 were set to about (2.4k, 10k) and Alpaca7B to (5.9k, 2.7k). $\epsilon$ was set to 0.25.
> Based on the representative NSEGen, FDR of GPT-3.5 was reported as (0.1561, 0.1478, 0.1146, fail, fail), and FDR of Alpaca7B was reported as (0.0251, 0.1478, 0.1146, fail, fail), respectively.
> * Two models show different FDR-Es on the NQ dataset. Considering this fact with the results, although Alpaca7B's $|\mathbf{Z}_U|, |\mathbf{Z}_E|$ on NQ dataset are larger, when the size is reduced, Alpaca7B fails to bound earlier than GPT-3.5, where other methods show similar trends.
> * Thus, for a stable bound, the certain size of $|\mathbf{Z}_E|$ seems to be needed proportional to the FDR(which is dependent on the model and data).
> * We will add this ablation studies by using plots in Appendix of our final manuscript.

---

### Official Review · Reviewer_abKP · 2024-07-12

**Soundness:** 3
**Presentation:** 2
**Contribution:** 2
**Rating:** 3
**Confidence:** 3

**Summary:**

This work presents a method for selective generation from language models for question answering. Their approach functions as a secondary decision function on top of an existing language model, determining whether to accept the language model's generation or to abstain. Their method is approach is based on constructing an entailment set of correct answers to a given question. The entailment set is defined as all answers that imply or textually entail the ground truth answer. They then design their secondary decision function such that it satisfies bounds on the correctness of accepted, generated answers based on whether it is within the constructed entailment set, or the answer not being in the ground truth entailment set. The authors then demonstrate that their method satisfies correctness guarantees based on conformal prediction, and demonstrate the efficacy of using their entailment set approach to identifying correctness empircally through experiments on open-domain question answering (NaturalQuestions) and with two language models (Alapaca7b and GPT3.5)

**Strengths:**

The goal of this work, selective generation for language models to prevent hallucinations and erroneous outputs is sounds and well motivated.

This work provides theoretical guarantees on the efficacy of the selective generation approach, builds upon insights from prior works on the efficacy of using textual entailment to aid in calibrating QA predictions.

**Weaknesses:**

Relying on single-directional textual entailment as a method for determining correctness of an answer is susceptible to accepting generations that produce that untrue fact or hallucinations in addition to the correct answer. [1] is a very relevant work (that was not cited) that similarly uses entailment to determine equivalence between two answers; however, in their work they rely on bi-directional entailment to ensure answer equivalence.

The baselines used in this work lack clear descriptions, and are not in line with current simple and effective methods for selective generation and calibration of language models [1, 2, 3]. [2] and [3] both do not require additional entailment training data and [1] takes a related approach using entailment to determine answer equivalence among a set of generated answers. [1] also does not require domain-specific entailment data.

[1] Semantic Uncertainty: Linguistic Invariances for Uncertainty Estimation in Natural Language Generation
Lorenz Kuhn, Yarin Gal, Sebastian Farquhar
ICLR 2023

[2] Self-consistency improves chain of thought reasoning in language models.
X. Wang, J. Wei, D. Schuurmans, Q. Le, E. Chi, S. Narang, A. Chowdhery, and D. Zhou.
ICLR 2023

[3] Language Models (Mostly) Know What They Know
Saurav Kadavath, Tom Conerly, Amanda Askell, Tom Henighan, Dawn Drain, Ethan Perez, Nicholas Schiefer, Zac Hatfield-Dodds, Nova DasSarma, Eli Tran-Johnson, Scott Johnston, Sheer El-Showk, Andy Jones, Nelson Elhage, Tristan Hume, Anna Chen, Yuntao Bai, Sam Bowman, Stanislav Fort, Deep Ganguli, Danny Hernandez, Josh Jacobson, Jackson Kernion, Shauna Kravec, Liane Lovitt, Kamal Ndousse, Catherine Olsson, Sam Ringer, Dario Amodei, Tom Brown, Jack Clark, Nicholas Joseph, Ben Mann, Sam McCandlish, Chris Olah, Jared Kaplan
ArXiv 2022

**Questions:**

The presentation of the experimental details could benefit from additional explanation. Clearly defining what the baselines are and how their implemented (not just the algorithms in the appendix) would greatly improve readability. This applies to the experimental results and metrics as well.

Evaluations could be designed in a way that (1) more clearly describe each methods efficacy and performance in the end task (QA in this case) and (2) allow for direct comparison against other methods also designed for calibration and selective generation.

**Limitations:**

The authors address that they do not report statistical significance.

---

> ### Author Rebuttal · Authors · 2024-08-06
>
> > Relying on single-directional textual entailment as a method for determining correctness of an answer is susceptible to accepting generations that produce that untrue fact or hallucinations in addition to the correct answer...
> - We agree that the bi-directional entailment is the best choice in terms of evaluating the "equivalence" of two sequences. However, for open-ended QA problem, GLM is prompted to generate a sequence in a free-form that often includes extra explanation. Therefore, considering the single-direction entailment as a correctness metric may be preferred.
> - Furthermore, the problem set-up that our algorithm considers is general since it can take any type of entailment $R$ as a correctness metric (Eq. (1)). Taking paraphrase generation as example, it is appropriate to let $R$ be the bi-directional entailment, and FDR-E is defined as $P${$G(x)\notin E_{\text{true}}(y)\wedge y\notin E_{\text{true}}(G(x))$}. Then, the implementation of our algorithm is exactly the same as we did with the open-ended QA problem.
> - Additionally, we conducted experiments using bi-directional entailment and compared them with the result from single-directional entailment in the open-ended QA problem using proxy labels, showing similar results. For GPT3.5, NSEGen can bound FDR-E to 0.1659 (where efficiency is 0.9804) when using single-directional entailment, while to 0.1594 (where efficiency is 0.9653) using bi-directional entailment. Other baselines showed similar trends.
>
> > The baselines used in this work are not in line with current simple and effective methods ... [R1, R2, R3].
> - We appreciate constructive suggestions on related work. Suggested papers [R1-R3] are mainly to design better uncertainty measures for the generated sequence (=a good “scoring function”). It is a very important issue in the NLP literature, and the efficiency of our algorithm implicitly depends on a given scoring function. However, “given” a scoring function, what our algorithm does is to learn a selective generator which refuses to generate the sequence based on a scoring function value. Therefore, designing a scoring function is out of our scope. [R1-R3] can be used to define the scoring function, and our algorithm provides a theoretical guarantee on FDR-E, irrespective of the choice of scoring functions.
>
> > [R2] and [R3] both do not require additional entailment training data and [R1] takes a related approach using entailment to determine answer equivalence among a set of generated answers. [R1] also does not require domain-specific entailment data.
> - We agree. But we also want to highlight that an extra entailment-labeled training set is necessary to “guarantee” the rate of hallucinations (Theorem 1).
> - Besides, it is also true that [R1] does not require “domain-specific” entailment data, where the transformer-based entailment classifier that [R1] utilizes is trained on the NLI dataset. While the classifier can be applied to arbitrary pairs of sequences, it underperforms on some language generation problems where the pair comes from different distribution.  For such cases, extra calibration data are needed to train an entailment classifier from scratch [R4], or to finetune the base model [R5].
>    * [R4] Yu Gui, Ying Jin, and Zhimei Ren. “Conformal Alignment: Knowing When to Trust Foundation Models with Guarantees.” ArXiv, 2024.
>    * [R5] Christopher Mohri, et al. “Learning to Reject with a Fixed Predictor: Application to Decontextualization.” ICLR, 2024.
>
> > The baselines used in this work lack clear descriptions…
>
> > Clearly defining what the baselines are and how they are implemented ... would greatly improve readability. This applies to the experimental results and metrics as well.
> - We clarified descriptions on baselines (SG-EM, SG-EL, SG-ES) along with extra baselines (SG-ES-Sup, SG-PL, SG-PFL) for clear comparisons.
>    * Supervised Baselines
>       * SG-EM: A supervised method that uses exact match metric
>       * SG-EL: A supervised method that uses semantic correctness encoded in the entailment label instead of exact match
>       * SG-ES-Sup: A method that uses same entailment set algorithm in SG-ES, where $Z_U = \emptyset$
>    * Semi-Supervised Baselines
>       * SG-PL & SG-PFL: They exploit the unlabeled data by pseudo-labeling entailment based on a threshold. Both algorithms are heuristics, since they do not control the mislabeling error.
>    * Metrics
>       * FDR-E refers to the ratio of whether a generated answer does not entail the true answer for a test sample.
>       * Efficiency refers to the ratio of data selected in the test set.
>
> > The presentation of the experimental details could benefit from additional explanation.
> - We employ two large language models (LLMs), GPT-3.5-Turbo and Alpaca7B. We create a dataset on textual entailment using the Natural Questions (NQ) dataset [43]. Based on the transformation method by [44], we convert the (q, a) pair into a declarative form for each model. Besides, semi-supervised learning algorithms use only 75% of the labeled data compared to what is used by supervised methods.
> - To control an FDR-E, we use two user-specified parameters $(\epsilon, \delta)$, where we use $(0.25, 0.02)$ unless specified. For our methods (i.e., SG-ES, NSEGen, and SG-ES-Sup), we have four parameters ($\epsilon_S, \epsilon_E, \delta_S, \delta_E$) which are mapped as follows: $\epsilon_S = \epsilon$, $\epsilon_E = 10^{-4}$, $\delta_S = \delta/2, \delta_E = \delta/2$. For other methods without using entailment sets, we use $\epsilon$ and $\delta$ accordingly.
>
> > Evaluations could be designed in a way that (1) more clearly describe each method's efficacy and performance in the end task (QA in this case) and (2) allow for direct comparison against other methods also designed for calibration and selective generation.
> - We have compared methods in attached files (Table 1, 2).
>
> > The authors address that they do not report statistical significance.
> - See Figure 1 in attached files.

---

> ### Author Response · Authors · 2024-08-13
>
> Dear Reviewer abKP,
>
> We again appreciate your constructive feedback to improve our submission. As the discussion period ends soon, we'd like to hear more about your opinion on our submission to address your concerns. Thanks!
>
>
> Best,
> Authors

---

### Official Review · Reviewer_7eme · 2024-07-15

**Soundness:** 3
**Presentation:** 3
**Contribution:** 3
**Rating:** 7
**Confidence:** 3

**Summary:**

The paper looks at a selective generative language system, meaning one which can produce a I-Don't-Know label rather than an answer, and calibrating it such that some guarantees on the precision of the system can be made.
The paper expands upon citation [1] on selective generation, to improve the efficiency by not requiring an exact match metric on the answer in order to be able to say whether it is correct or not. This is important for language generation tasks where there are endless possibly valid answers. It does this via the use of entailment measures, ie saying whether a candidate question+answer pair have positive/negative or neutral entailment.
The paper reports theoretical analysis of when the proposed method can be learnt within given error margins, doing so under reasonable assumptions of the data being IID. This may not truly be the case for say actual question-answer datasets, but is a required assumption for the theoretical statements that are presented.
The paper also presents empirical results on a question-answer dataset (Natural Questions), comparing against other selective generation methods with exact matching (rather than entailment), and with entailment sets (from labels or inferred).
The proposed method which learns the selective threshold is shown to have the best false-discovery-rate (ie the inverse of precision on the generated answer).

**Strengths:**

The requirement of having labels for answer correctness is clearly a large limiting factor in language generation tasks given the large number of possible correct answers. Relaxing this via entailment is therefore well motivated.
Thorough theoretical analysis given, albeit under

**Weaknesses:**

* Say more about what language problems this could and could not be applied to. The key appears to be in being able to have an accurate entailment function. For how many types of language-generation problems is such obtainable?

* Some data on rates of IDK generation on the test set would be useful to see.

* Minor suggestion: an algorithm box showing the steps for the whole proposed approach would be helpful for the reader.

* The paper would benefit from an editor / proofreading. It doesn't impact understanding, but there are several small typos (lines 34, 69, 367 as some examples)

**Questions:**

* In figure 2, are the SG-EM and SG-ES systems scoring 0? I'm not familiar with the prior literature, but with this paper being strongly influenced by [1] where SG-EM is from, it seems surprising these 2 baselines are so poor.

**Limitations:**

* The theoretical analysis relies on the IID assumption, which won't hold on real world language datasets. This is noted by the authors.
* Not clear how many language problems this is applicable to.

---

> ### Author Rebuttal · Authors · 2024-08-06
>
> > Say more about what language problems this could and could not be applied to. The key appears to be in being able to have an accurate entailment function. For how many types of language-generation problems are such obtainable?
> Thanks for raising a nice point. We expect our method can be generalized to a variety of language generation problems, but we currently consider open-ended QA due to the limitation of the entailment classifier that we rely on.
> * In particular, as you have mentioned, the key point is whether we have access to an accurate entailment function for the given language generation problem. Specifically, as can be seen in Algorithm 1, the accuracy of the entailment function depends on the performance of the entailment classifier. Therefore, the applicability of our algorithm depends on the choice of the entailment classifier and its quality on the specific type of language-generation problem we consider. In this paper, we consider the transformer-based textual entailment classifier which is originally trained on the NLI dataset. Since the NLI dataset consists of pairs of premises and hypotheses in a declarative form of moderate length, the entailment classifier would not perform well on pairs of long sequences (e.g., summarization). For such cases where the pretrained entailment classifier provides inaccurate predictions, extra data are needed to train an entailment classifier from scratch [R1], or to finetune the base model [R2].
>     * [R1] Yu Gui, Ying Jin, and Zhimei Ren. “Conformal Alignment: Knowing When to Trust Foundation Models with Guarantees.” ArXiv, 2024.
>     * [R2] Christopher Mohri, et al. “Learning to Reject with a Fixed Predictor: Application to Decontextualization.” ICLR, 2024.
>
> > Some data on rates of IDK generation on the test set would be useful to see.
> * Rates of IDK generation correspond to 1 - efficiency, which refers to the ratio of data selected in the test set. Efficiency is reported by figures and tables in our response.pdf (or the paper).
> * If you want some data examples of IDK generation, we sampled three examples of GPT3.5 that NSEGen says IDK, in addition to Table 2 of the paper.
> * Question: who sang there she was walking down the street
>     * Correct Answer: Manfred Mann
>     * Generated Answer: The Beatles sang “There she was walking down the street”
> * Question: how long is the ferry ride from cape may to lewes
>     * Correct Answer: 80 minutes
>     * Generated Answer: The ferry ride from Cape May to Lewes is approximately 1 hour and 15 minutes.
> * Question: what is the first book in the hive series
>     * Correct Answer: Institute of Villainous Education
>     * Generated Answer: The first book in the Hive series is “The Hatching” by Ezekiel Boone.
>
> > Minor suggestion: an algorithm box showing the steps for the whole proposed approach would be helpful for the reader.
> * Thank you for your suggestion. We will add the algorithm box on the final manuscript.
>
> > The paper would benefit from an editor / proofreading. It doesn't impact understanding, but there are several small typos (lines 34, 69, 367 as some examples)
> * Thank you for your suggestion. We will proofread over the draft and fix typos and errors for better readability.
>
> > In figure 2, are the SG-EM and SG-ES systems scoring 0? I'm not familiar with the prior literature, but with this paper being strongly influenced by [1] where SG-EM is from, it seems surprising these 2 baselines are so poor.
> * The main reason for the small FDR result of SG-EM is that it does not consider semantic correctness of answers (i.e., it only considers the answer is correct only if it is exactly the same as a given true answer). Thus, to achieve a desired FDR rate, [1] finds a conservative selective generator (i.e., $\tau$ is close to one thus mostly saying IDK), resulting in low FDR to achieve a desired FDR rate.
> The main reason for the small FDR result of SG-ES is due to the not-performant scoring function, i.e., a scoring function $f_{M_1}$ usually does not have high scores on correct answers. In this case, Algorithm 3 does not tightly achieve a desired FDR rate, it finds a conservative selective generator (i.e., $\tau$ is close to one thus mostly saying IDK), resulting in low FDR to achieve a desired FDR rate (as the SG-EM case).
> * The reason why the score is exactly '0' is because we did not report the scores on the figure when the methods cannot be bound under $\epsilon_S$, for the reasons above. (In this case, $\tau$ is the largest value in the calibration data which is ~= 1).
> * We will add this analysis in the paper and to avoid such confusion, we will use Table to annotate the cases where $\epsilon_S$ guarantees are not satisfied.

---

> > ### Author Response · Authors · 2024-08-10
> >
> > BTW, we have a formatting issue in our first answer. Our response starts from "Thanks for raising..." in the question quote. Please consider this when you read our response. Thanks!

---

### Author Rebuttal · Authors · 2024-08-06

We appreciate reviewers’ valuable feedback and constructive comments. In this global response, we delineate the structure of our responses.
* We provide individual responses to questions of each reviewer.
* We provide pdf, including requested experiment results.
* We also suggested an improved method due to a request – We can provide a proof (also a slight variation of the original proof) on the correctness of the improved method if required.

---

### Decision · Program_Chairs · 2024-09-25

**Decision:**

Accept (spotlight)

**Comment:**

The paper presents an entailment-based false discovery rate by leveraging logical entailment to define the relationship between true and generated answers for GLMs. They introduce a supervised selective generation approach using entailment labels to control the FDR.  The paper also reports theoretical analysis of when the proposed method can be learnt within given error margins.  Any major concerns from reviewers seem resolved in the rebuttal/discussion.